# Monoclonal antibody levels and protection from COVID-19

Eva Stadler [1], Martin T. Burgess[2], Timothy E. Schlub [3], Shanchita R. Khan [1], Khai Li Chai[4], Zoe K. McQuilten[4,5], Erica M. Wood [4,5], Mark N. Polizzotto[6,7], Stephen J. Kent [8,9], Deborah Cromer[1], Miles P. Davenport [1]✉ & David S. Khoury [1]✉

Multiple monoclonal antibodies have been shown to be effective for both prophylaxis and therapy for SARS-CoV-2 infection. Here we aggregate data from randomized controlled trials assessing the use of monoclonal antibodies (mAb) in preventing symptomatic SARS-CoV-2 infection. We use data on the in vivo concentration of mAb and the associated protection from COVID-19 over time to model the dose-response relationship of mAb for prophylaxis. We estimate that 50% protection from COVID-19 is achieved with a mAb concentration of 96-fold of the in vitro IC50 (95% CI: 32−285). This relationship provides a tool for predicting the prophylactic efficacy of new mAb and against SARS-CoV-2 variants. Finally, we compare the relationship between neutralization titer and protection from COVID-19 after either mAb treatment or vaccination. We find no significant difference between the 50% protective titer for mAb and vaccination, although sample sizes limited the power to detect a difference.

Vaccination has been shown to be highly effective at preventing both symptomatic and severe COVID-19 (reviewed by Cromer et al.[1]). However, vaccination is less effective in many immune-compromised and elderly individuals where immunogenicity and clinical data show considerably impaired responses to vaccination[2,3]. Multiple monoclonal antibody products have been shown to be effective as pre- and post-exposure prophylaxis against pre-Omicron variants[4−6], as well as when administered therapeutically early in infection[7−11]. We recently analyzed the available data on antibody treatment of symptomatic SARS-CoV-2 infection to determine the dose-response relationship between the antibody dose administered (after conversion to a neutralizing dose equivalent) and the protection from progression to

hospitalization[12]. However, the dose-response curve for monoclonal antibody administration as prophylaxis of COVID-19 has not yet been determined. Here we adopt an alternative approach, comparing the loss of antibody in vivo with the loss of efficacy of monoclonal antibodies over time following administration. In addition, we use data on the loss of neutralization and protection observed to new variants to inform this relationship[13]. Using this data on temporal changes in monoclonal antibody concentration and changes in potency to new variants, we estimate the relationship between in vivo antibody concentration and protection, which may provide a valuable clinical tool for predicting the efficacy of new monoclonal products and existing products against new variants[12]. Finally, we assess whether neutralizing

[1]Kirby Institute, University of New South Wales, Sydney, NSW, Australia. [2]School of Mathematics and Statistics, University of New South Wales, Sydney, NSW, Australia. [3]Sydney School of Public Health, Faculty of Medicine and Health, University of Sydney, Sydney, NSW, Australia. [4]Transfusion Research Unit, School of Public Health and Preventive Medicine, Monash University, Melbourne, VIC, Australia. [5]Department of Clinical Haematology, Monash Health, Clayton, VIC, Australia. [6]Clinical Hub for Interventional Research, College of Health and Medicine, The Australian National University, Canberra, ACT, Australia. [7]Department of Clinical Haematology, Canberra Region Cancer Centre, The Canberra Hospital, Canberra, ACT, Australia. [8]Department of Microbiology and Immunology, University of Melbourne at the Peter Doherty Institute for Infection and Immunity, Melbourne, VIC, Australia. [9]Melbourne Sexual Health Centre and Department of Infectious Diseases, Alfred Hospital and Central Clinical School, Monash University, Melbourne, VIC, Australia. ✉e-mail: m.davenport@unsw.edu.au; dkhoury@kirby.unsw.edu.au

antibodies mediate protection or merely correlate with protection by comparing the relationship between neutralization titer and protection after vaccination[14] and in naïve individuals receiving monoclonal antibodies. Together this work provides a quantitative framework for dissecting the mechanisms of protection in vaccination and informing the use of critical immunotherapies.

## Results

### Aggregating studies of monoclonal antibodies as prophylaxis

We searched MEDLINE, PubMed, Embase, and the Cochrane COVID-19 Study Register for randomized placebo-controlled trials of SARS-CoV-2-specific monoclonal antibodies (mAbs) used as pre-exposure and peri-exposure prophylaxis for COVID-19. We included only studies where both protection from symptomatic infection and pharmacokinetic information of the monoclonal antibody were provided within the same study. We identified six eligible studies assessing monoclonal antibodies as pre-exposure and peri-exposure prophylaxis for COVID-19[4–6,15,16]. The antibodies used in these studies were casirivimab/imdevimab (three studies), bamlanivimab, cilgavimab/tixagevimab, and adintrevimab. One of these studies did not provide data on the pharmacokinetics of the antibody (bamlanivimab)[16] and was excluded. Of the remaining five studies, three reported a break-down of cases in treatment and control arms by time since administration, and two studies had data on the timing of cases that could be extracted from

the publication[4–6,13,15] (Table S1). Four of these five studies assessed protection at a time before the Omicron variants were the dominant circulating variants (Table S2). One study assessed protection in two time periods; firstly in a pre-Omicron period when the Delta variant was the dominant circulating variant, and separately later when Omicron variants BA.1 and BA.1.1 were the dominant variants[13]. The overall efficacies against pre-Omicron variants in the included studies ranged from 68.6% to 92.4%. We identified a trend for lower efficacies with increasing time since administration and against the escaped variant, the latter being reported previously by Schmidt et al.[13] (Fig. 1).

### A significant dose-response relationship between protection and mAb concentration

To investigate whether declining efficacy with time and new variants were indicative of a dose-response relationship between mAb concentration and efficacy, we compared the antibody concentrations reported within different time intervals in each study with the reported efficacy at the corresponding time interval (details of time intervals used in analysis provided in Table S1). We found that when we only considered studies where the predominant circulating variant was a non-VOC (i.e. excluding Schmidt et al.[13] which analyzed adintrevimab protection from the Delta and Omicron variants), there was a significant correlation between antibody concentration and efficacy ($RR = 0.39$ per $\log_{10}$-increase in antibody concentration, $p = 0.003$,

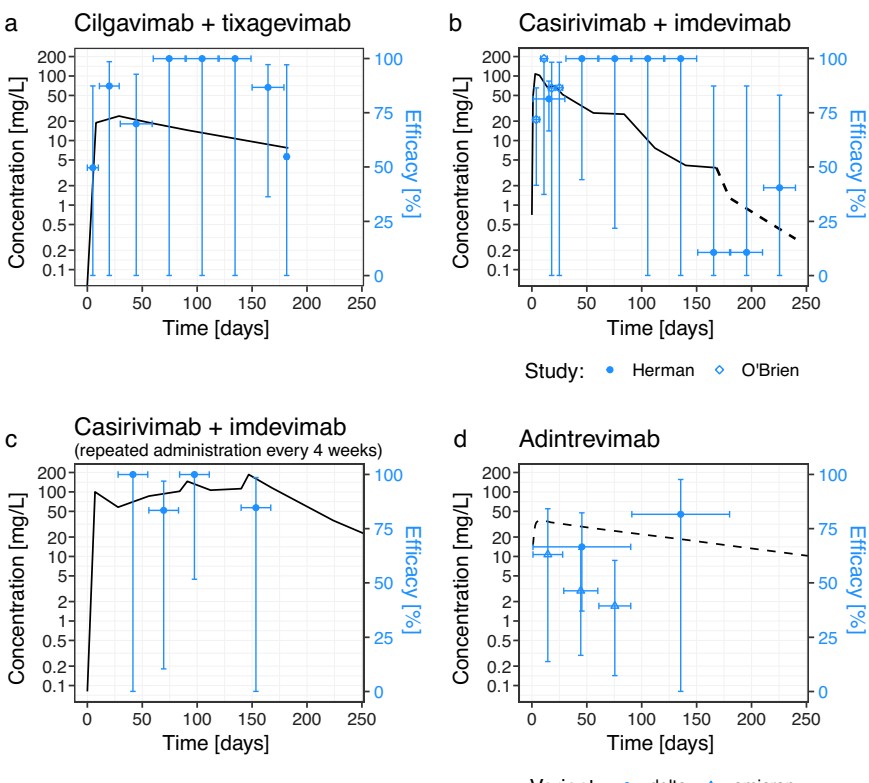

**Fig. 1 | Reported protection and antibody concentration from RCTs of monoclonal antibodies in preventing COVID-19.** The efficacy at each time interval is shown in blue (points indicate observed efficacy, horizontal error bars indicate time interval and vertical error bars represent 95% CIs of efficacy). The antibody concentration is shown in black. **a** Antibody concentration ($n = 1776$ individuals) and efficacy data ($n = 5172$ individuals) for cilgavimab/tixagevimab was extracted from Levin et al.[4] **b** Single administration of casirivimab/imdevimab data are a combination of data from O'Brien et al.[6] and Herman et al.[15] who report on the same clinical trial over different follow-up intervals. Efficacy data were reported weekly over the first four weeks in O'Brien et al. (diamonds) ($n = 1505$), and monthly for eight months in Herman et al. (circles) ($n = 1683$). Antibody concentration data was reported up to day 168 in O'Brien et al. (solid line, **b** $n = 12$), and modeling of the pharmacokinetic profile of the antibody concentration, reported in Herman et al., was used to inform the antibody concentration between 168 and 240 days (dashed line, **b**). **c** Isa et al.[5] reported efficacy ($n = 969$) and in vivo concentration after repeated administration of 1.2 g of casirivimab/imdevimab every 4 weeks ($n = 723$). Hence, the antibody concentration did not decline as in the other studies. **d** The modeled concentration of adintrevimab after a single administration was extracted from the study by Schmidt et al.[13] The efficacy of adintrevimab was reported both when the delta variant was dominant (circles) ($n = 1267$) and when Omicron variants BA.1 and BA.1.1 were dominant (triangles) ($n = 378$).

generalized linear mixed model (GLMM) and chi-squared test). However, a significant association between antibody concentration and efficacy was lost when we included data on efficacy against the Omicron variant from ref. 13 (RR = 0.69, p = 0.13). This is likely due to the loss of neutralizing potency of adintrevimab against the Omicron variant (Table S3), thus lower efficacy would be expected (for a given antibody concentration) against these escaped variants.

To adjust for the different neutralizing potencies of each antibody and loss of potency against different variants, we normalized antibody concentration using the in vitro IC50 for each antibody against the dominant variant circulating at the time of the study (Table S2). These in vitro IC50 for different antibody/variant combinations were obtained from a meta-analysis of in vitro studies (Table S3 and Figure S1, using data from the Stanford University Coronavirus and Resistance Database[17]). We found that after normalizing by the in vitro IC50, we found a significant relationship between efficacy and mAb concentration (as a fold of the in vitro IC50) (RR = 0.40 per $\log_{10}$-increase in antibody concentration, p < 0.0001). Together this suggests that in vivo monoclonal antibody concentrations adjusted by neutralizing potency are correlated with efficacy to prevent COVID-19.

To test the robustness of this correlation we performed sensitivity analyses. Firstly, our analysis uses a combination of data from true pre-exposure prophylaxis settings and also from settings of peri-exposure prophylaxis (i.e., in individuals after some degree of known contact with a COVID-19 index case). Thus, we repeated the analysis using only those studies where true pre-exposure prophylaxis was assessed and found the relationship remained significant (RR = 0.45 per $\log_{10}$-increase in antibody concentration, p < 0.0001). Further, in leave-one-out, leave-two-out, and so on analyses, we found this relationship remained significant in all cases except when both the Schmidt et al.[13] and Herman et al.[15] studies were omitted (Table S4), suggesting a sensitivity of the results to these two studies.

We next fitted this dose-response relationship with a range of functional forms, including a threshold model, an exponential risk model, and a generalized logistic model among others (Table S5). We found that all these models produced very similar qualities of fit as assessed by the Akaike Information Criterion (AIC) (Table S5, maximum AIC differences ≤3.5, Figure S2). We chose the best fitting model (lowest AIC), which is the logistic function with a maximum efficacy of 100%, and was also the same model used previously to describe the relationship between neutralizing antibodies and efficacy for vaccination[14]. Fitting this logistic dose-response relationship to the data, we estimate that a concentration of 96-fold the in vitro IC50 (95% CI: 32–285) is associated with 50% efficacy (Table S6, Pearson's goodness-of-fit test, $\chi_{22}$ = 19.1, p = 0.64, Fig. 2). Against the ancestral virus, this equates to a plasma antibody concentration of 0.41 mg/L concentration for cilgavimab/tixagevimab, 0.40 mg/L for casirivimab/imdevimab, and 0.53 mg/L for adintrevimab being required for 50% protection against COVID-19. We found this model fit was robust to uncertainty in the antibody concentration and IC50 estimates from the meta-analysis (tested using a bootstrapping approach, see Supplementary Methods "Multiple imputation of mAb concentration and in vitro IC50 data" and Table S7). Together, this shows that monoclonal antibody concentration, once normalized for in vitro neutralizing potency, is significantly associated with protection from symptomatic COVID-19 infection.

**Predicting monoclonal antibody efficacy against new variants**
A major challenge in the COVID-19 pandemic has been in decision-making around whether pharmaceuticals shown to be effective against ancestral SARS-CoV-2 should continue to be used when new variants emerge. This is especially true for monoclonal antibody therapies, where recommendations on the use of mAb therapeutics have changed numerous times with the emergence of each Omicron subvariant[18,19]. This has been particularly difficult when a mAb loses

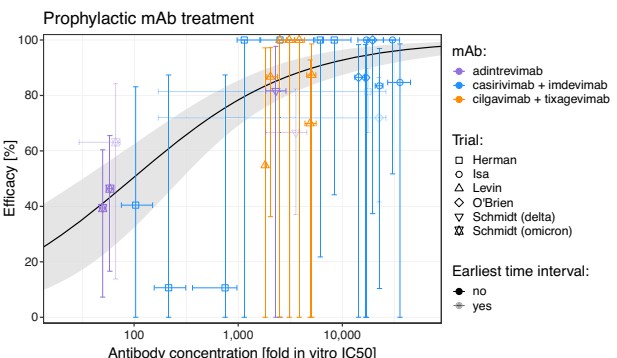

**Fig. 2 | Dose-response relationship between antibody concentration and protection.** The estimated geometric mean antibody concentration and protective efficacy in the matching study and time interval (expressed as a fold of the in vitro 50% inhibitory concentration (IC50) of each antibody) are shown. Horizontal error bars indicate the maximum and minimum (mean) antibody concentrations observed during each time interval, and vertical error bars indicate the 95% confidence interval of the efficacy. We estimate a dose-response relationship (black line) by fitting a logistic model with maximum 1 (i.e., maximal efficacy 100%) to the data and estimate the 95% confidence region using parametric bootstrapping (gray shading) (n = 24 individual observations). The best-fit parameters of the dose-response relationship are: 50% efficacy with an antibody concentration of 96.2-fold in vitro IC50 (95% CI: 32.4–285.2) and a slope parameter of 1.3 (95% CI: 0.9–1.8). Efficacy data reported early after treatment (i.e., in the first time point reported in the study) were excluded from the model fitting (low opacity data points), since antibody concentration changed rapidly over this time interval and to ensure exclusion of unidentified infections that might have occurred before treatment.

partial, but not complete, recognition of a new variant. Given the ongoing development of novel broadly cross-reactive monoclonal antibodies for the prevention and treatment of COVID-19 (e.g., trial ID: NCT05648110, https://clinicaltrials.gov/), this remains an important question. If we assume that the relationship defined here between antibody concentration (normalized to in vitro neutralizing IC50) and efficacy will continue to hold for different variants of concern, as it has for vaccine-induced neutralizing antibodies[1,20], we can use the dose-response relationship in Fig. 2 to estimate the loss of efficacy and duration of protection of monoclonal antibodies to new variants. For example, cilgavimab/tixagevimab administered intramuscularly at a dose of 300 mg is predicted to maintain >50% protection for 581 days (95% CI: 433–730 days) against the ancestral variant, since the in vitro IC50 is 4.27 ng/mL and the half-life of this antibody combination is 95 days (Fig. 3, Tables S8, Figure S3). However, given the in vitro IC50 increases 8.9-fold to Omicron BA.2, it is predicted that this same dose would provide protection above 50% for 282 days (95% CI: 133–430 days) against this variant. In this example, this mAb combination would need to be administered every 282 days in order to maintain at least 50% efficacy against Omicron BA.2. Importantly, cilgavimab/tixagevimab is not predicted to attain 50% efficacy against Omicron BA.1 even shortly after treatment (because of the large increase in the in vitro IC50 to this variant (Table S3)). Similar estimates can be determined from this analysis for the other mAbs based on the in vitro IC50 of these mAbs to different SARS-CoV-2 variants (Fig. 3).

Another formulation of this question is to ask "What is the maximum increase of IC50 (drop in neutralization titer) that can be tolerated while still maintaining a minimum duration of protection?". For example, if we wish to provide a period of at least 30 days with >50% protection, then cilgavimab/tixagevimab, casirivimab/imdevimab, and adintrevimab, at the current doses, can tolerate at most 56.5-fold (95% CI: 19.1–167.4), 143.8-fold (95% CI: 48.5–426.2) and 61.1-fold (95% CI: 20.6–181.3) increases in in vitro IC50 compared to the in vitro IC50 against the ancestral variant, respectively. Figure 3d shows the predicted duration of >50% protection for casirivimab/imdevimab,

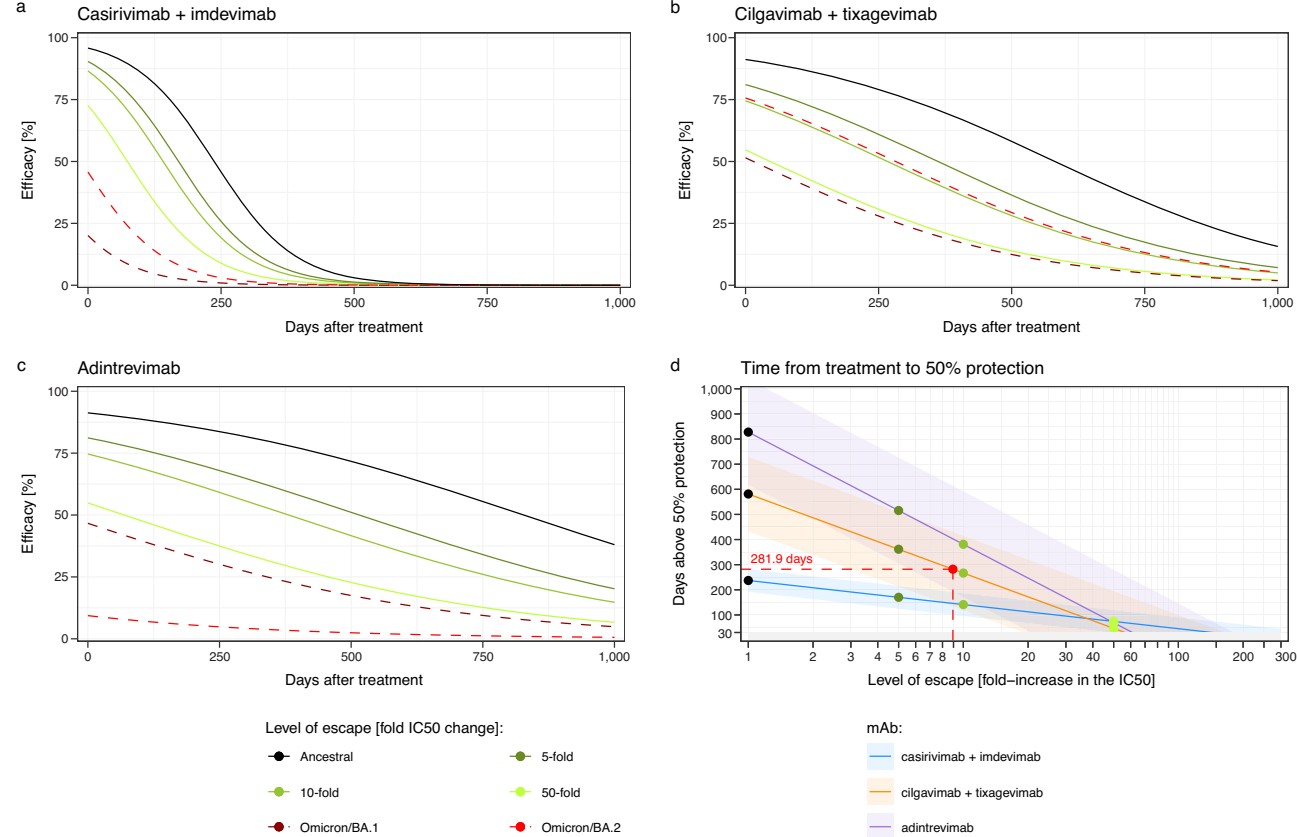

**Fig. 3 | Duration of protection against SARS-CoV-2 variants.** Using the dose-response relationship of antibody concentration and protective efficacy in Fig. 2 and the estimated half-life of antibodies after treatment (28.9 days (95% CI: 26.6–31.6) for casirivimab/imdevimab, 94.7 days (95% CI: 83.5–109.4) for cilgavi-mab/tixagevimab and 134.4 days (95% CI: 132.6–136.3) for adintrevimab, see Figure S3 and Table S8), we predict the efficacy over time for these antibody combinations (black line, **a**–**c**). We also estimate the protection over time of these antibody combinations given different fold-increases in the 50% inhibitory concentration (IC50), which may be experienced toward new variants (colored lines).

**d** For each hypothetical loss of potency of these antibodies (i.e., fold increase in IC50, *x* axis), we predict the number of days each antibody will remain above 50% protection. The shaded regions indicate the 95% confidence interval of the duration of protection (using the 95% CI for 50% protection: 32.4–285.2). We note that casirivimab/imdevimab, cilgavimab/tixagevimab and adintrevimab are predicted to tolerate up to a 143.8-, 56.5- and 61.1-fold drop in potency to a new variant (compared to the ancestral variant), respectively, and still be expected to maintain 30 days of >50% protective efficacy.

cilgavimab/tixagevimab, and adintrevimab for any given fold-change in in vitro IC50. Using this analysis, we see that all of these mAb are predicted to be ineffective against at least some of the recent circulating Omicron subvariants (e.g., BA.2.75, BQ.1.1 and XBB, where data on IC50 shift is available), because of larger shifts in the IC50 (Table S3). Going forward, this analysis will provide drug developers with a means of using in vitro neutralization data to predict the efficacy of candidate broadly neutralizing mAb against novel SARS-CoV-2 variants, as well as to guide dosing/dosing interval decisions for promising monoclonal antibodies in order to achieve a specified level of protection against the most relevant circulating variants.

**Comparing monoclonal antibody prophylaxis with vaccine-induced protection**

Multiple lines of evidence have established that neutralizing antibody titers correlate with protection from COVID-19 in vaccinated individuals[14,21–23]. An important question is whether neutralizing antibodies are mechanistic in mediating this protection, or merely correlate with protection[24]. Similarly, if antibodies are able to directly mediate protection, identifying the magnitude of their contribution to overall protection (compared to other mechanisms) is an important question. One way to address this is to compare the level of protection achieved after administration of antibodies alone with that achieved after vaccination. Antibody administration alone should reflect the antibody-related contribution to protection, while vaccination should

incorporate both antibody- and cell-mediated protection. Recently, Schmidt et al.[13] reported the loss of in vitro neutralization of adin-trevimab to Omicron BA.1 and BA.1.1 resulted in a corresponding loss of efficacy that was consistent with the relationship between reports of vaccine effectiveness. Here we address this question by integrating the available data across currently available studies on monoclonal antibody prophylaxis. We then used an established correlate of vaccine protection for COVID-19[14] to analyze whether prophylaxis against COVID-19 after passive antibody administration is achieved at similar levels of neutralization to protection observed after vaccination.

Figure 4a compares the efficacy of high-potency mRNA vaccines[25,26] and monoclonal antibody prophylaxis in the prevention of symptomatic SARS-CoV-2 infection (with the ancestral or delta variant) in the first 3 months post-vaccination or administration of antibody. We find that the observed mean risk reduction achieved with mAbs was significantly lower than that achieved by mRNA vaccination (RR = 2.87, 95% CI: 1.48–5.56, *p* = 0.002, Wald test). This difference corresponds to a mean efficacy of 84.8% (95% CI: 76.0–90.8) from mAbs and 94.5% (95% CI: 91.6–96.7) from mRNA vaccination. However, this does not take into account the neutralizing antibody titers in the different groups. Thus, we next compared the level of protection achieved for a given neutralizing antibody titer after either vaccination (from Khoury et al.[14]) or after treatment with monoclonal antibodies (Fig. 4b). To compare titers between vaccination and monoclonal antibody administration, we normalized the titers of each to a scale

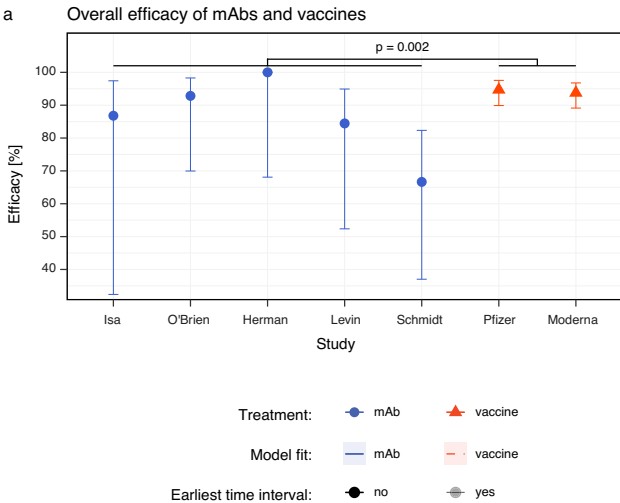

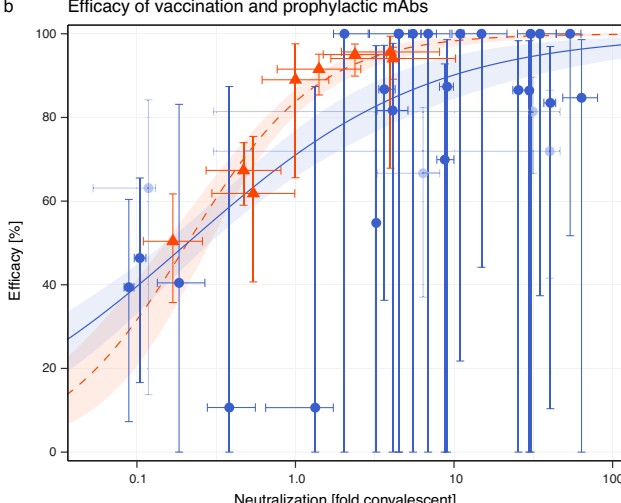

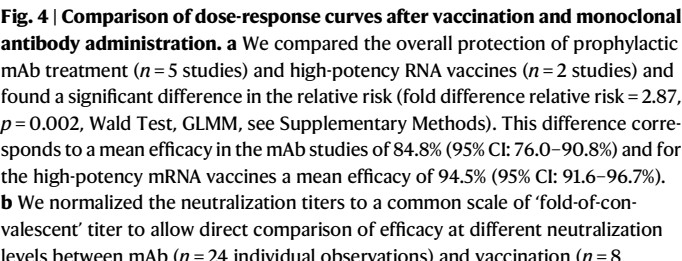

**Fig. 4 | Comparison of dose-response curves after vaccination and monoclonal antibody administration. a** We compared the overall protection of prophylactic mAb treatment (*n* = 5 studies) and high-potency RNA vaccines (*n* = 2 studies) and found a significant difference in the relative risk (fold difference relative risk = 2.87, *p* = 0.002, Wald Test, GLMM, see Supplementary Methods). This difference corresponds to a mean efficacy in the mAb studies of 84.8% (95% CI: 76.0–90.8%) and for the high-potency mRNA vaccines a mean efficacy of 94.5% (95% CI: 91.6–96.7%). **b** We normalized the neutralization titers to a common scale of 'fold-of-convalescent' titer to allow direct comparison of efficacy at different neutralization levels between mAb (*n* = 24 individual observations) and vaccination (*n* = 8

individual data points). Here we show the best fitting model (lines) to the data (points) was a model where the slope is allowed to vary between mAb studies (blue/solid line) and vaccine studies (red/dashed line) but the neutralization titers giving 50% protection is equal for mAb prophylaxis and vaccination (based on model comparisons with the likelihood ratio test, see Figure S4, Figure S5, and Table S9). Shaded regions indicate the 95% confidence regions of the fitted model.
**a, b** Vertical error bars indicate the 95% CI of the efficacy, and horizontal error bars indicate the maximum and minimum (mean) neutralizing antibody titer observed during each time interval (blue) or 95% CI of the mean neutralizing antibody titer (red).

relative to the geometric mean titer of neutralizing antibodies seen in convalescent individuals (neutralization titers against ancestral virus in individuals after infection in the first wave of COVID-19). We term this the 'fold-convalescent' scale (as previously described for the studies of vaccination[14], and as described in the Methods for mAbs). We used a model comparison approach in order to test whether the neutralizing antibody titers associated with a given level of protection for vaccines and mAbs were consistent or different (fitting models with the same or different parameters for vaccine and mAb and comparing fits using a likelihood ratio test and AICs, see Supplementary Methods, Table S9, Figure S4, and Figure S5). There was no evidence for a difference in the neutralization titer required for 50% protection between vaccination and mAb treatment (fold-change in titer for 50% protection in mAb compared to vaccination is 0.81, 95% CI: 0.26–2.51, Figure S4). Given the limited statistical power, these results show that if a difference between these groups exist, the fold difference is unlikely to be lower than 0.26 or higher than 2.5. Further, our analysis showed that the best-fit model was one where the same dose-response relationship existed for both vaccination and mAbs but with the estimated slope being higher for vaccination (Fig. 4b, Table S9). Together, these results indicate that similar levels of neutralizing antibodies from either vaccination or the administration of monoclonal antibody are associated with 50% protection from COVID-19, but with a trend towards a lower protection achieved with monoclonal antibodies compared to vaccination at high neutralizing antibody titers (Fig. 4 and Figure S5).

## Discussion

Here we demonstrate a relationship between the monoclonal antibody concentration and efficacy in preventing COVID-19. Further, we estimate the concentration of antibody required to have a high confidence of maintaining at least 50% protection. Our model fitting enabled us to quantify the uncertainty in this relationship and estimate that if a treated population can maintain a mean in vivo monoclonal antibody concentration of >96-fold (95% CI: 32–285) of the in vitro IC50 of the antibody to the circulating variant, they should maintain >50% efficacy

against COVID-19. Analysis of the dose-response curve for monoclonal antibodies allows prediction of the level and duration of protection against different SARS-CoV-2 variants (Fig. 3). Our results suggest that casirivimab/imdevimab, cilgavimab/tixagevimab, and adintrevimab would provide >50% protection against the ancestral SARS-CoV-2 strain for 586, 581, and 548 days respectively. Unfortunately, this analysis also predicts that these mAb prophylaxis regimens will have lost protection against most of the more recent circulating variants because the change in IC50 to the variants is considerably greater than the thresholds determined here (Fig. 3d, Table S3). This supports the decisions of regulators who have now withdrawn recommendations for all mAbs for use as prophylaxis[27]. We suggest that the analysis presented here will provide a tool for ongoing development of broadly neutralizing monoclonal antibodies, allowing in vitro data to be used to inform dosing choices of candidates in trials.

Counterintuitively, although antibodies with a longer half-life are expected in general to provide protection for longer, these are also expected to lose more 'days of protection' for a given fold increase in IC50 (to a new variant), compared to mAbs with shorter half-lives (Fig. 3). The higher susceptibility of therapeutics with longer half-lives to fold-shifts in the IC50 has been discussed previously for antimalarial products[28], and can be explained by considering that when antibodies lose 2-fold neutralization against a new variant it is equivalent to the mAb losing one half-life of time above a threshold. Therefore, for a 2-fold increase in IC50 a mAb with a 100-day half-life will lose 100 days above a specified threshold, whereas a mAb with only a 30-day half-life will lose 30 days above the same threshold.

The estimated in vivo concentration of antibody required for 50% protection from COVID-19 is much higher than the level of antibody required to neutralize virus in vitro (~100-fold), suggesting that in vivo neutralization may be much less efficient than the observed neutralization in vitro. This difference between in vitro IC50 and the in vivo 50% protective titer is not unexpected, given the major differences between infection in these environments. For example, in vitro neutralization assays usually involve pre-incubation of antibody and virus

for an hour before exposure to cells. Similarly, the in vitro IC50 in plaque reduction assays estimates the antibody concentration required to neutralize 50% of virions. However, the dose required to completely neutralize large inocula may be considerably higher[29]. In addition, in vivo antibody titers are assessed in the serum. However, antibody concentration at the mucosa is lower than the plasma level[30], and thus higher (serum) titers may be required to achieve neutralization on mucosal surfaces.

We and others have previously shown that neutralizing antibodies are a correlate of protection from COVID-19[14,21,23,31,32]. A major question in understanding vaccine-mediated immunity is whether neutralizing antibodies are simply a surrogate marker of protection or are mechanistic in protecting individuals from symptomatic infection[14]. To-date it has only been possible to consider this question indirectly. For example, we have noted that the drop in neutralizing antibodies against new variants and over time both provide good predictions of the change in efficacy of vaccines over time and against new variants[1,20]. Schmidt et al. recently showed that the monoclonal antibody adintrevimab lost neutralization to Omicron BA.1 and BA.1.1, and a corresponding loss of prophylactic efficacy compared with that predicted by the relationship between neutralization titer and vaccine effectiveness studies[13]. Here, we extracted data on antibody PK and temporal changes in efficacy from five monoclonal antibody studies (including from Schmidt et al.[13]), and compared this with an established immune correlate of protection after vaccination that has been validated across a number of settings[1,14,20,33,34]. This allows a direct comparison of the protection provided by passively administered monoclonal antibodies versus vaccine-induced polyclonal antibodies. We have been able to show that the neutralization titer required for 50% protection by vaccination or monoclonal antibodies is comparable (Table S9), although a predicted trend towards higher efficacy of vaccines compared to mAb at high neutralizing antibody titers was observed (Fig. 4b and Figure S5). Importantly, the power to detect a difference in these comparisons is limited by the data available.

The difference in protection at a given neutralization titer between vaccination and monoclonal antibody therapy may be due to the additional benefit in vaccinees of a polyclonal antibody response, other non-neutralizing functions of antibodies, recall of immune memory, and/or other cellular immune responses. In particular, adintrevimab, tixagevimab, and cilgavimab all have modified Fc domains to increase the antibody half-life[13,35], and tixagevimab and cilgavimab have additional mutations to reduce Fc-related antibody functions. These functions may contribute to the estimated trend towards higher protection for vaccines at high neutralizing titers. While our analysis has shown that neutralizing antibodies alone are sufficient to provide a high level of protection from COVID-19 at the neutralization titers induced by vaccination, it is not possible to conclude from this analysis that neutralizing antibodies are necessary for protection. Also, since adintrevimab, casirivimab, and imdevimab have seemingly intact Fc receptor functions, differences in efficacy between these products may arise from other non-neutralizing functions that we have not directly considered here. We note that evidence in animal models supports the findings that neutralizing antibodies mediate protective immunity[36], with some showing an additional benefit of Fc receptor function[37].

The analysis presented here has a number of limitations. Firstly, our dose-response analysis requires comparison of the in vivo measured antibody concentrations and the estimated in vitro IC50s. This relied on a meta-regression of estimates of the IC50 of each mAb against different variants, and we observed here a large between-study variation (Figure S1). This means that the results of any particular study or assay may vary quite considerably from the central estimate of the IC50 for analysis of all studies. Despite the limitations of the IC50 meta-analysis, we found that the fitted model for the relationship between antibody concentration and efficacy was robust to the uncertainties in

the IC50s (Table S7 and Supplementary Methods). Further, when restricting the analysis to only studies conducted when non-VOC predominated, unadjusted antibody concentrations were similarly predictive of efficacy.

An additional limitation is that we did not have access to raw data from the clinical studies. Data were requested from all corresponding authors of the original studies (31 March 2023) but were not provided by the time of revision (30 June 2023), thus we relied on extraction of data from the published reports. This involved manual extraction of data from figures in some cases, which carries implicit risks of error. Data were extracted independently by two authors, and the results were compared to resolve discrepancies[38]. Further, we are reliant upon population level rather than individual-level data on antibody concentration, half-lives, and clinical outcomes broken down by time intervals as reported in, or extracted from, the published studies. Thus, we could not account for between-subject variability, or subjects lost to follow-up (although fortunately, these numbers are relatively small). Additionally, the analysis is strongly influenced by the results from the Herman et al.[15] and Schmidt et al.[13] studies, given their contribution of data at lower effective antibody concentrations (Fig. 2). It is also evident that each study data point on efficacy contained considerable uncertainty (with wide confidence intervals), this contributed somewhat to wide confidence intervals in the overall fitted model (Fig. 2). To gain more precise estimates of the dose-response curve for the use of monoclonal antibodies for prophylaxis, combining the results from more studies, preferably with individual-level data available, and with longer follow-up times (where this is ethical) would be helpful.

Other differences between the design of the included studies may have impacted the analysis, which we could not account for directly. For example, seropositivity was zero at baseline in all studies except Isa et al.[5], and vaccination of participants after treatment was allowed in Herman et al.[15] (Table S2). Fortunately, in our sensitivity analysis, we found that the correlation between antibody correlation and efficacy was robust to removing either of these studies (Table S4). Also, uptake of vaccination in Herman et al. was similar between treated and control arms (~35%). This likely lowered the power of the study to detect efficacy at later time points, but otherwise was not expected to have a large impact on the efficacy estimates in this study (Table S2). In addition, studies used different modes of delivery of the monoclonal antibodies, i.e., casirivimab/imdevimab was administered subcutaneously in the Isa[5], O'Brien[6], and Herman[15] studies, whereas cilgavimab/tixagevimab and adintrevimab were administered intramuscularly[4,13]. Plasma antibody concentrations appear to increase more slowly following intramuscular administration compared with subcutaneous administration of casirivimab/imdevimab (Fig. 1 and Table S8), and thus it is possible that there is a delay until protective antibody concentrations are achieved. To avoid this difference and also to account for the risk of infection around the time of antibody administration, we omitted the earliest time interval from our analysis (which encompassed the first 7, 10, 27, 28, 30, or 90 days across different studies, Table S1).

Finally, when comparing mAb prophylaxis and vaccination, there are some additional limitations. In particular, this comparison involves converting the antibody concentration data to a fold of convalescent scale. However, we have not shown that such normalization can account for all the differences between neutralization titers achieved by vaccination and mAb administration. For example, it is possible that the normalization employed here oversimplifies the comparison since it does not appreciate potential differences between neutralization readouts for polyclonal sera and monoclonal antibodies. Further, only a subset of studies ($n = 19$) in our meta-regression reported the geometric mean neutralization titer of a suitable panel of convalescent sera. In addition, the definition of convalescent sera was specified differently in each study, which introduced some potential confounders to these aggregated estimates. Even so, this approach has

revealed surprisingly similar prophylactic and vaccine efficacy for a given neutralization titer on the fold of the convalescent scale (within the statistical power of the data).

Our conclusion that monoclonal antibody therapy and vaccination provide similar protection at equivalent neutralizing antibody titers is consistent with a recent publication by Follmann et al.[39]. In that study the authors performed an analysis using simulated individual neutralizing antibody titers from one trial of casirivimab and imdevimab[6,15] (from the study in Fig. 1b here) and one mRNA-1273 vaccine trial[23,26]. The authors found that monoclonal antibodies could explain most of the observed vaccine protection at high antibody levels, although they note that such a comparison was not possible at low neutralizing titers due to limited power. Our analysis suggests that vaccination and monoclonal antibody administration also provide similar levels of protection at low antibody levels.

Vaccination has provided a high level of population immunity to COVID-19. However, there remain a number of subgroups in which vaccination is either not possible or ineffective (largely due to immunodeficiency). The use of monoclonal antibodies for prophylaxis in these cohorts has the potential to provide long-term protection from both symptomatic and severe COVID-19 for these vulnerable groups. However, the frequent observation of novel SARS-CoV-2 variants that escape antibody recognition has raised significant challenges in predicting monoclonal antibody protection against new variants. Further work is required to obtain more data on protection at low antibody levels, as well as to validate predictions of prophylactic efficacy against SARS-CoV-2 variants. Within this context, the work presented here provides a quantitative and evidence-based framework for predicting monoclonal antibody efficacy that can be used in the assessment of novel therapeutics or in designing optimal regimes for new SARS-CoV-2 variants.

## Methods

### Search strategy for studies of COVID-19 prophylaxis with monoclonal antibodies

As also detailed in[12], we used the results of the systematic review performed by the Cochrane Team's on monoclonal antibodies to prevent COVID-19[40]. Briefly, searches were performed in MEDLINE, PubMed, Embase, and the Cochrane COVID-19 Study Register from inception to 30 November 2022, for randomized controlled trials of monoclonal antibodies for the prevention of COVID-19. We identified six studies. We extended this systematic review from 1 January 2022 to 31 January 2023 using a shortened randomized controlled trial-only search on PubMed following the same search strategy and identified no new studies. Our search identified five studies where treatment efficacy and monoclonal antibody concentration were reported for the same cohort (Table S1 and Table S2). These studies were in a mixture of true pre-exposure prophylaxis and peri-exposure prophylaxis settings (Table S1). Data were extracted from these independently by two authors (ES and SRK), and discrepancies were resolved through discussion (Supplementary Methods).

Two studies, O'Brien et al.[6] and Herman et al.[15], reported results from the same clinical trial over different follow-up intervals (4 weeks and 8 months respectively). Thus, to avoid duplication of the same trial results, we integrated the results from these studies. In particular, the O'Brien trial reported outcomes on a weekly basis for 4 weeks whereas the Herman trial reported outcomes on a monthly basis for 8 months. Therefore, for these trials, the weekly outcomes reported in O'Brien were used for weeks 2–4 after administration (the initial week was omitted due to rising antibody levels in this period), and the results from Herman et al., were used for the months 2–8 only. In addition, antibody concentration data for the cohort was extracted from Figure S4 of O'Brien et al. from 0–168 days, whereas Herman reported only pharmacokinetic model predictions of the concentration over the interval of 30–240 days. Therefore, the raw O'Brien et al. antibody

concentration data were used from 0–168 days and the predicted concentrations from Herman et al. were used for the remaining interval, 168–240 days after treatment. This is indicated in Fig. 1b by a different line type (solid for in vivo concentration data from O'Brien et al. and dashed for modeled concentration data from Herman et al.). In Fig. 1d, the dashed line for the concentration data also denotes modeled concentration data rather than in vivo measurements of concentration.

### Estimation of antibody concentration on fold IC50 scale and estimate of neutralization titer on fold-convalescent scale

Antibody concentrations in each study were extracted and normalized as a ratio to the mean IC50 for that antibody (against the relevant variant, see Table S2) obtained from a meta-regression of the available data on antibody binding obtained from the Stanford University Coronavirus and Resistance Database (https://covdb.stanford.edu/)[17] (IC50 data and meta-regression are described in detail in the Supplementary Methods). The IC50s from this meta-regression are provided in Table S3. Note that since some of these studies used antibody combination therapies, the total antibody concentration (sum of both antibody components) was used for the antibody concentration. When comparing the protection observed in monoclonal antibody trials against the protection seen after vaccination, we converted the estimated neutralization titer in individuals treated with mAbs to a 'fold-of-convalescent' scale, since all vaccine trials had been aligned based on this scale previously[14]. This was estimated by taking the antibody concentration as a fold of the IC50 described here and dividing by the mean convalescent neutralization titer which was also estimated from a meta-regression of titers reported in the Stanford database (and described in detail in the Supplementary Methods).

### Dose-response fitting with maximum likelihood approach

To estimate the dose-response relationship between efficacy ($E$) and antibody concentration ($c$), we fitted a number of functional forms describing the relationship (all models are detailed in Table S5), and we selected the best-fitting model by the Akaike Information Criterion (AIC). We fitted all dose-response models to the data with a maximum likelihood approach (as previously described[12]). Briefly, the likelihood function used in this optimization was,

$$L(\mathbf{p},\mathbf{b}) = \prod_{\tau} \text{Binom}\left(e_c^{\tau}, n_c^{\tau}, b^{\tau}\right) \times \text{Binom}\left(e_t^{\tau}, n_t^{\tau}, b^{\tau} \times (1 - E(c(\tau)|\mathbf{p}))\right), \quad (1)$$

where $\mathbf{p}$ denotes a vector of parameters of the dose-response relationship (which differ by model, Table S5), and $\mathbf{b}$ is a vector of the baseline risk parameters, $b^{\tau}$, for each trial/time interval combination ($\tau$), and Binom is the probability mass function of the binomial distribution. For each trial/time interval combination, $\tau$, $e_c^{\tau}$ and $e_t^{\tau}$ are the numbers of events (symptomatic infections) in the control and treatment groups, respectively, and $n_c^{\tau}$ and $n_t^{\tau}$ are the total number of individuals in the control and treatment groups, respectively. The likelihood function assumes that the baseline risk of infection, $b^{\tau}$, is reduced by the efficacy of treatment for the treatment group. When fitting the model parameters, the initial guess of the baseline risk for each trial/time interval combination was $b^{\tau} = e_c^{\tau}/n_c^{\tau}$. The parameter $c(\tau)$ is the ($\log_{10}$) concentration of monoclonal antibodies (in the fold-IC50 scale) in the trial time interval $\tau$.

The negative log-transform of this likelihood function was minimized using the *nlm* optimizer in the R statistical software (Version 4.2.1) package to estimate the log-transform of the model parameters $\mathbf{p}$ and the baseline risk $\mathbf{b}$. The optimizer was run 100 times using randomly generated initial parameters for the $\mathbf{p}$ drawn uniformly from designated ranges (Table S5).

Model fitting was used for parameter estimation and hypothesis testing, the latter using a likelihood ratio test for nested models

(for vaccine and mAb comparison), and the AIC for non-nested models. In all fitting, we excluded the earliest time interval of each study to account for the rapid change of the antibody concentration over this time interval and ensure exclusion of unidentified infections that might have occurred immediately before treatment (see Table S1 for the time intervals included in the analysis).

The best-fitting model was a logistic function with maximum efficacy of 1, i.e., 100% (Figs. 2 and 4). Under this dose-response relationship, the efficacy for treatment with mAb concentration *c* is given by

$$E(c|k,c_{50}) = \frac{1}{1 + \exp(-k \times (\log_{10}(c) - \log_{10}(c_{50})))}, \quad (2)$$

where *k* is a slope parameter determining the steepness of the relationship and $c_{50}$ is the concentration that gives 50% efficacy.

The confidence intervals of the fitted model and the fitted model parameters were estimated by parametric bootstrapping. The covariance matrix of the fitted parameters was calculated from the inverse of the Hessian from the model fit[20] and used to compute the 95% confidence intervals of the estimated parameters. The log-transform of the model parameters was drawn randomly from a multinomial Gaussian distribution 100,000 times, using the covariance matrix (and the function *rmvnorm* from the package *mvtnorm* in the R statistic package (Version 4.2.1)[41]). The 95% confidence regions of the fitted models in Figs. 2 and 4b were estimated by evaluating the model at each neutralization titer using each of the 100,000 bootstrapped parameter estimates and taking the 2.5% and 97.5% percentiles of the evaluated models at each antibody concentration.

### Reporting summary
Further information on research design is available in the Nature Portfolio Reporting Summary linked to this article.

## Data availability
All data are available publicly on GitHub at https://github.com/david-s-khoury/COVID19-mAb-prophylaxis.

## Code availability
All code is available publicly on GitHub at https://github.com/david-s-khoury/COVID19-mAb-prophylaxis.

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

## Acknowledgements
This work is supported by Australian government Medical Research Future Fund awards GNT2002073 (to M.P.D., S.J.K.), MRF2005544 (to S.J.K., M.P.D.), MRF2005760 (to M.P.D.), MRF2016062 (to S.J.K., M.P.D., D.S.K.) an NHMRC program grant GNT1149990 (S.J.K. and M.P.D.), and the Victorian Government (S.J.K.). NHMRC Investigator grants 1173528 (to D.C.), 1194811 (to Z.K.M.), 1194811 (to E.M.W.), 2016491 (to S.J.K.), and 1173027 (to M.P.D.). K.L.C. is supported by Ph.D. scholarship from the Leukemia Foundation, the Hematology Society of Australia and New Zealand (HSANZ), and Monash University. The Melbourne WHO Collaborating Center for Reference and Research on Influenza is supported by the Australian Government Department of Health.

## Author contributions
E.S., M.N.P., S.J.K., D.C., M.P.D., and D.S.K. conceived the study. E.S., S.R.K. K.L.C., Z.K.M., E.M.W., M.P.D., and D.S.K. contributed to the curation of the data. E.S., M.T.B., T.E.S., M.P.D., and D.S.K. contributed to the formal analysis and visualization. All authors contributed to the writing of the manuscript. All authors approved the final manuscript.

## Competing interests
M.N.P. declares receiving provision of drugs for clinical trials from CSL Behring, Takeda, Grifols, Emergent Biosciences, and Gilead. Z.K.M. and E.M.W. declare research funding from CSL Behring to Monash University for work outside this project. The authors declare no other competing interests.

## Ethics
This work was approved under the UNSW Sydney Human Research Ethics Committee (approval HC200242).
