## [Peer Review File · Nature Communications]

Reviewers' Comments:

Reviewer #1:

Remarks to the Author:

Positive feedback.

I was impressed by the data hunt performed by the authors to find and digitize relevant data from the literature. Analyses are relatively well described; the approach used (likelihood-based) is a solid choice for this type of problems (although variability in IC50 calls for a Bayesian approach). I also liked the attempt to normalize efficacies of different antibodies by IC50 (even though I don't believe we have evidence it is appropriate). The way to compare immunity by nAbs and vaccine is appropriate statistically (but the data may be too noisy to make a more rigorous conclusion). I also liked the idea to use the results from the analyses to predict duration of protection (although I would do it differently, to be more relevant to clinic).

Major concerns

1. My main concern is with quality of the digitized data. Presumably the data in graph are correct but we don't know for sure, and authors may have made errors in that process. I think for studies of high importance to human health it is inappropriate to use a sloppy approach like this to "digitize" data. Authors must contact the writers of relevant papers on clinical trials and use those data. In addition, I found inappropriate the point that "data will be provided upon publication". NO. This is important study, so authors must provide data with the submission, so reviewers can examine those - sometimes graphical representation of the data is not very good - e.g., Figure 2 is very noisy with too many error bars that make it impossible to interpret any fit (i.e., any model may be able to fit such data).
2. I understand and appreciate the usefulness of comparing different nAbs in terms of efficacy to prevent COVID19 caused by different variants. The approach to use IC50 seems interesting but I am not sure I can believe IC50 provided in the paper. First, variability in provided individual values for IC50 (Fig S1) is too large to be useful to normalize any other study. Second, it is unclear if IC50 calculations cited are comparable at all. WHO spent some time to come up with proper way to standardize infection assays and how neutralization is measured. What happens if authors focus on one Ab and do not use IC50 (perhaps restrict the data to one SARS-CoV2 variant). In any case, importance of IC50 variability must be incorporated into the analyses, e.g., perhaps using Bayesian statistics. In addition, Fig 2 must have some error in x axis due to IC50 variability.
3. nAb dynamics is not given per patient. Obviously, this is important to predict how a given individual responds to the nAb therapy (protected or not). Fig 1 data must be improved to include as much as possible of individual Ab dynamics, and quantified using population-based approaches, e.g., mixed effects.
4. Only one model for Ab protection is considered (line 391 but equations are not numbered!). Why is this the right model? What about alternative models that may be consistent with the data? For example, a threshold model may be reasonable, or hill-like function (e.g., PMID: 32453765). A note - estimate for k includes 1, so it seems that the model in which $k=1$ should fit the data fine and is a simpler model.
5. How do we know that the model fits the data well? The presented data are very noisy so perhaps the model does fit the data. Providing some statistical evidence of model fit quality is needed - e.g., using chi-square test?

6. When comparing protection by nAbs and the vaccine authors must investigate i) what sample size would be required to detect a given difference in vaccine vs. nAb efficacy (power analysis)? ii) how do errors in estimating IC50 may contribute to not detecting a difference between two efficacy estimates?

7. Authors need to perform some type of uncertainty analysis for all major claims to find conditions under which these break. For example, statement that nAbs and vaccine-induced protection are similar - how wrong can you be?

Minor concerns

All equations must be numbered and referred to in the text by numbers.

What happens if you do not normalize the data by IC50? Does the analysis fail?

It is unclear in how graphs in Fig 3 are useful. Are we concerned with a particular level of efficacy? Should one actually show how often Abs need to be injected to maintain a level of efficacy? Also, scales for x axes in 1-c should be the same.

line 136 - half-life of this Ab seems longer than the typical 2-3wks. Why?

Line 237-240: I think this will confuse a lot of people. I would suggest more about practical aspects - e.g., if we want to protect a person from SARS-CoV2, we would need to inject a given Ab at dose X every day Y to maintain Z level of protection. That people will understand.

It is nice that authors list limitations. But there are more of these - see comments above about the data in particular.

Line 326 - 328 - authors provide no evidence that this approach mitigates systemic biases in the analysis.

Figure 1 must have same scales for all panels for better interpretation.

Line 112 (supplement): The 95% CIs for efficacy are likely to be correct only approximately. Perhaps using resampling/bootstrap is a better approach.

Table S3 - leave one out. What about leaving 2 out? 3 out? This will verify robustness of the conclusions.

Table S4 - efficacy cannot be more than 100%!

Table S7 - how relevant is peak nAb concentration? Seems that the whole kinetics should be more important than the peak. Please investigate.

Table S6 - numbers look weird, references seem like a "power".

Reviewer #2:

Remarks to the Author:

Thank you for inviting me to review this manuscript. This work follows on from previous analyses by the same authors on the dose-response relationship between MAb dose after conversion to neutralizing dose equivalence, and protection from hospitalisation when MAb given early after symptomatic SARS-CoV-2 infection. The use of the dose-response relationship between normalized

IC50 and efficacy to predict the clinical efficacy of a new variant against existing MABs is particularly important and informative. The authors aggregate data from 5 MAB RCTs with data on clinical efficacy alongside PK data, to a range of MABs.

This manuscript has considerable practical implications for how existing and future MABs are authorised/continued to be used in treatment moving forwards.

I only have minor comments and suggestions:

1. Please state whether every participant in each of the studies had samples taken for PK data or just a subset.
2. Figure 1 - add 'every 4 weeks' to 1c 'repeated administration' so clear by just looking at figure when the doses would be
3. For generation of data in figure 2, state in main text that the timeframes/windows of efficacy and antibody concentration for each study used are found in Table S1
4. Data on antibody efficacy vs IC50 - it isn't clear in the results or figure 2 legend whether the X axis data (i.e. fold in vitro IC50) were generated using normalised IC50 to multiple variants or just to ancestral virus. To include Schmidt et al data (where efficacy against Delta and Omicron are what is available) I assume it may be the former. If IC50 fold against multiple variants was amalgamated, how did the authors deal with the issue of IC50 above the limit of the in vitro assay used in the case of multiple omicron-MAB combinations? What would make most sense is to only include IC50 fold data for viruses that were circulating at the time any particular study was conducted (so matched to the efficacy data). This may be what the authors did but it needs clarifying.
5. Some discussion of how some design aspects of the primary studies may have affected the estimates would be useful:
 - a. To what extent did they recruit seronegative participants only (and therefore the efficacy can be attributed solely to the MAB)
 - b. To what extent did they allow vaccination to occur or include vaccinated participants? This is important in considering the efficacy retained in the face of lower MAB antibody concentrations, and introduce greater uncertainty to the estimates
6. In previous publications, the authors have elegantly used a fold-convalescent approach to normalise data from several studies and estimate the relationship between in vitro Nab titre and clinical efficacy. The same approach is used here to compare mAb and vaccine efficacy and I appreciate the value of doing so. Do the authors have data that would allow them to go further and standardise their output (at least for some MAB against ancestral virus) to WHO standard 20/136 IU50/mL by estimating the relationship between antibody concentration and neutralisation titre? Or the newer 21/338 (variant) standards? It would help with comparison with other studies reporting on correlates of protection in IU50/mL.

Reviewer #3:

Remarks to the Author:

In this manuscript, Stadler and colleagues extract data on protection from symptomatic SARS-CoV-2 infection and antibody pharmacokinetics from clinical trials of monoclonal neutralizing antibodies used for pre- and peri-exposure prophylaxis. Based on this data, they model a relationship between antibody concentrations relative to in vitro IC50s and estimated protective efficacy, including longitudinally for individual antibodies/antibody combinations. Finally, they model antibody efficacy for neutralizing activity relative to neutralization seen in convalescent individuals and compare this data

to similar data previously determined for vaccinees.

The analysis is overall valuable because it attempts to identify mAb-mediated serum neutralization titers that provide protection from symptomatic COVID-19 when administered as prophylaxis. Particularly in individuals with insufficient responses to vaccination, the results can support the development of strategies for antibody prophylaxis (e.g., dosing, frequency of dosing, adjustments for newly emerging variants).

Limitations and suggestions:

- Because of the overall limited number of cases, the efficacy confidence intervals for individual time periods are very wide (often ranging from almost 0% to 100% for the peer-reviewed data) (Figure 1; Figure 4). This applies as well to the overall predicted IC50 for 50% protection (>1 log range in the CI). This imprecision could be emphasized somewhat more in the limitations section.

- Many cases/breakthrough infections are derived from the Schmidt et al. study on adintrevimab, which is – as of yet – not published in peer-reviewed form (i.e., published data may change).

- Data on the casirivimab/imdevimab combination were collected both in a pre- and in a peri-exposure setting, whereas for the other antibodies, only the pre-exposure setting was investigated. Because these are different clinical scenarios, they cannot necessarily be compared. It seems more prudent to only analyze the pre-exposure setting.

- In the Herman et al. study of prolonged follow-up after a single casirivimab/imdevimab injection, participants were eligible for COVID-19 vaccination, and 35% in the antibody group received at least one vaccination during follow-up (similar number in the placebo group) and there were almost no more symptomatic infections after vaccination. Did the authors account for this?

- Efficacies of mAb administration and vaccination are compared by analyzing neutralization titers relative to those seen in convalescent individuals. For the vaccinees, previously published data from Khoury et al, Nat Med 2021, were used. This data was based on studies in which vaccinated individuals and convalescent individuals were tested side-by-side (all live virus). In contrast, for the assessment of mAb efficacy as related to fold-convalescent neutralization, the data for convalescent neutralization were derived from a wider range of neutralization studies (refs 7-17 in supplement). These studies were not related to the monoclonal antibody infusion trials and used more diverse assays (pseudovirus vs. live virus; focus reduction vs. luminescence; etc.) Neutralization assays can differ in the effects of individual antibody contributions (e.g., pronounced activity for RBD-targeting neutralizing antibodies – such as the monoclonal antibodies tested here – in assays using ACE2-overexpressing cells). The authors discuss assay differences in the context of mAbs (lines 318 ff.). However, they should also comment on how the comparison between vaccinated and mAb-treated individuals in Figure 4 might be affected (curve for vaccinated is based on comparison of polyclonal vs. polyclonal antibodies; curve for antibody group based on neutralization assays using RBD-targeting antibodies vs. polyclonal antibodies).

- In Figure 4B, very little information below 0.1-fold-convalescent neutralization is available (none at all for the vaccinees). In the original Khoury et al., Nat Med 2021, paper, the curve for vaccinees is cut-off at 0.1. It might be prudent to do so here as well (show curves up to the lowest available data point).

More minor:

- Figures 4B and S3 appear to be identical.

- In Figure S1, serum neutralization titers from convalescent individuals (in blue) are shown with the

same y-axis as monoclonal antibodies. However, this data is probably ID50s (dilutions) rather than concentrations. This should be corrected/not shown with the same y-axis.

- In addition to neutralization, antibodies may act through Fc-mediated effector function. Compared to casirivimab/imdevimab and adintrevimab, tixagevimab and cilgavimab have been Fc-modified for reduced Fc interactions. In contrast, half-life extending mutations have been introduced into adintrevimab as well as tixagevimab and cilgavimab; these mutations may also affect FcRn-mediated mucosal deposition. These differences may complicate the comparison of different antibodies and could be briefly discussed.

REVIEWER COMMENTS

Reviewer #1 (Remarks to the Author):

Positive feedback.

I was impressed by the data hunt performed by the authors to find and digitize relevant data from the literature. Analyses are relatively well described; the approach used (likelihood-based) is a solid choice for this type of problems (although variability in IC50 calls for a Bayesian approach). I also liked the attempt to normalize efficacies of different antibodies by IC50 (even though I don't believe we have evidence it is appropriate). The way to compare immunity by nAbs and vaccine is appropriate statistically (but the data may be too noisy to make a more rigorous conclusion). I also liked the idea to use the results from the analyses to predict duration of protection (although I would do it differently, to be more relevant to clinic).

We thank the reviewer for their positive comments.

Major concerns

1. My main concern is with quality of the digitized data. Presumably the data in graph are correct but we don't know for sure, and authors may have made errors in that process. I think for studies of high importance to human health it is inappropriate to use a sloppy approach like this to "digitize" data. Authors must contact the writers of relevant papers on clinical trials and use those data.

We thank the reviewer for their comments on data extraction and agree that for studies of high importance such as this it would be ideal if raw data were freely available / provided by the original authors. Unfortunately, this is not currently always the case, particularly where potential commercial interests are involved (as for these antibody products). E.g. adintrevimab has not received approval for use unlike the other antibodies considered here, and thus a commercial interest exists in the use of the data from their studies. Requests for data are frequently ignored, or legal reasons cited that prohibit sharing. In this case we wrote to all authors of the studies used in our analysis on 31 March 2023 and as of the time of resubmission we have not yet received any of the requested data for a range of reasons including:

- No response received to our request (for two of the studies)
- Response that legal approval is being sort and indication this is not likely (for one study)
- Response that they will seek advice internally to determine if data can be provided – but no subsequent response (two studies).

On the other hand, we also note that digital data extraction using WebPlotDigitiser is specified as an acceptable means of data extraction within the Cochrane handbook for systematic reviews (<https://training.cochrane.org/handbook/current/chapter-05>). This guide specifies that extraction should be performed by two independent scientists, and accordingly we have ensured all data was extracted independently by two of the authors.

We compared the extracted viral load data and found strong agreement in extracted values (all differences between extractions were $<0.11\text{-log}_{10}$ (i.e. $<1.4\text{-fold}$), see below figure comparing extracted values from two extractors), and the geometric mean of extracted data was used. For the clinical outcome data, this was also extracted independently by two authors, with identical results across all data, except for two instances where discrepancies were resolved through discussion, and agreement by all extractors that an error had occurred by one extractor. This brings our data acquisition methods in line with that expected by Cochrane systematic review guidelines. We believe that the revised extraction protocol is consistent with the standards of the field.

In addition, I found inappropriate the point that "data will be provided upon publication". NO. This is important study, so authors must provide data with the submission, so reviewers can examine those - sometimes graphical representation of the data is not very good - e.g., Figure 2 is very noisy with too many error bars that make it impossible to interpret any fit (i.e., any model may be able to fit such data).

We agree providing data and code for reviewers is necessary and apologise it was not included in the original submission. We have now provided a private link to the GitHub repository for the reviewers and editorial team (<https://gitfront.io/r/user-7600629/ydG4sSGDRYWe/COVID19-mAb-prophylaxis/>), which will be made public after publication.

2. I understand and appreciate the usefulness of comparing different nAbs in terms of efficacy to prevent COVID19 caused by different variants. The approach to use IC50 seems interesting but I am not sure I can believe IC50 provided in the paper. First, variability in provided individual values for IC50 (Fig S1) is too large to be useful to normalize any other study.

We thank the reviewer for their comments concerning the IC50 for each antibody here and below. We agree with the reviewer that determining the sensitivity of the analysis reported

here on the IC50 meta-analysis and inherent uncertainty in that analysis is important. However, we would like to clarify a misconception, that although the study heterogeneity is large (as noted by the reviewer), the central estimate of the IC50's across the field has much greater certainty. In part because of the large number of studies and capturing the study/assay dependent differences with random effects in the model. Though again, we agree with the reviewer that testing the sensitivity of the results to the IC50 meta-regression is important and we have now:

- 1) updated the source of IC50 data to the more comprehensive Stanford database (<https://covdb.stanford.edu/>).
- 2) performed sensitivity analysis to determine whether using IC50 to calibrate the Ab concentration data or using concentration data alone without adjustment yield similar results.
- 3) refitted the model to account for uncertainty in the x-axis using a bootstrapping approach with Rubin's rules¹.

We found that these offered strong validation of our results (detailed below). In particular, expanding the in vitro IC50 meta-analysis to use data from the Stanford database has meant that we have vastly more IC50 data for each antibody and for many more COVID-19 variants (including more recent variants BQ.1.1 and XBB). We used a linear mixed effects model with censoring (lme4 package) of IC50 above 10,000 ng/ml to estimate the IC50 for each mAb and variant combination, and we used a random effect to account for study/assay differences.

We note that the primary concern of the reviewer was whether the variability of the IC50 in the original study reduced the reliability of the results. It is important to emphasise that we have now used a completely different database to estimate the IC50 in the revised version of the manuscript. This had no effect on the overall conclusions of the study, reinforcing the robustness of the conclusions.

Second, it is unclear if IC50 calculations cited are comparable at all. WHO spent some time to come up with proper way to standardize infection assays and how neutralization is measured. What happens if authors focus on one Ab and do not use IC50 (perhaps restrict the data to one SARS-CoV2 variant).

We agree with the reviewer that efforts have been made to determine methods to standardize neutralisation assays, and these have relied on the use of calibration samples. Unfortunately, as we have described previously², these standards have proven very limited in their ability to truly calibrate data from different assays. After calibrating assays based on WHO standards inter-assay variability in neutralising titres still span 50-fold (as stated in the WHO's own document on the standards – ref ³).

Even so, we agree with the reviewer that testing the sensitivity of our conclusions to our normalisation by the in vitro IC50 is an important validation. We have now tested whether the relationship between antibody concentration (without adjusting for in vitro IC50) is correlated with efficacy across these studies. We find the correlation is significant when we exclude the data for Omicron variants (i.e. excluding data from Schmidt et al., RR=0.39 per log₁₀-increase in antibody concentration, p=0.003). However, the correlation is not

significant when we add in the Schmidt study (RR=0.69, p=0.13). The reason for this is clear, as the IC50 of adintrevimab differs around 100-fold between the wild-type and Omicron variants (Supplementary table S3). Therefore ignoring the IC-50 (assuming that the same concentration works equally well for both variants) leads to a prediction of much higher efficacy against the Omicron variant. After adjusting for the IC50, the relationship is highly significant (even with the Schmidt et al. study). We feel this additional analysis is important and provides some evidence to support the need for IC50 normalisation and that it is more relevant when there are big differences in IC50 (such as to new variants). This analysis is now included in the manuscript and we have now discussed these limitations more explicitly in the discussion:

Line 104: To investigate whether declining efficacy with time and new variants was indicative of a dose-response relationship between mAb concentration and efficacy, we compared the antibody concentrations reported within different time intervals in each study with the reported efficacy at the corresponding time interval (details of time intervals used in analysis provided in Table S1). We found that when we only considered studies where the predominant circulating variant was a non-VOC (i.e. excluding Schmidt et al.⁴ which analysed adintrevimab protection from the Delta and Omicron variants), there was a significant correlation between antibody concentration and efficacy (RR=0.39 per log₁₀-increase in antibody concentration, p=0.003, generalized linear mixed model (GLMM) and chi-squared test). However, a significant association between antibody concentration and efficacy was lost when we included data on efficacy against the Omicron variant from⁴ (RR=0.69, p=0.13). This is likely due to the loss of neutralising potency of adintrevimab against the Omicron variant (Table S3), thus lower efficacy would be expected (for a given antibody concentration) against these escaped variants.

To adjust for the different neutralising potencies of each antibody and loss of potency against different variants, we normalized antibody concentration using the in vitro IC50 for each antibody against the dominant variant circulating at the time of the study (Table S2). These in vitro IC50 for different antibody / variant combinations were obtained from a meta-analysis of in vitro studies (Table S3 and Figure S1, using data from the Stanford University Coronavirus and Resistance Database⁵). We found that after normalising by the in vitro IC50, we found a significant relationship between efficacy and mAb concentration (as a fold of the in vitro IC50) (RR=0.40 per log₁₀-increase in antibody concentration, p<0.0001). Together this suggests that in vivo monoclonal antibody concentrations adjusted by neutralising potency is correlated with efficacy to prevent COVID-19.

In any case, importance of IC50 variability must be incorporated into the analyses, e.g., perhaps using Bayesian statistics. In addition, Fig 2 must have some error in x axis due to IC50 variability.

We found this suggested analysis by the reviewer particularly useful for testing the robustness of the model fitting. We have now re-run our model fitting (presented in Fig 2) using a bootstrapping approach and Rubin's rules¹, to account for the uncertainty in the x-

position. Briefly this involved first randomly sampling the x-position of each data point in Figure 2, by sampling the uncertainty in the timing of the sample, and the associated uncertainty in the Ab concentration at the time of sampling. Secondly, we sampled from the uncertainty in the IC50 value from our meta-regression. The model fitting procedure was then repeated on each bootstrapped dataset, and Rubin's rules were applied to estimate the within-bootstrap uncertainty (which captures the uncertainty in the y-axis) and between-bootstrap uncertainty (which captures the uncertainty in the x-axis). We found that the fitted model taking into account the x-position uncertainty using this bootstrapping approach gave nearly identical parameter estimates and uncertainty to the model fit with the simplified approach of assuming no x-position uncertainty. This was because the within bootstrap uncertainty was far greater than the between bootstrap uncertainty. This is perhaps counterintuitive, but the result arises because, despite the large degree of inter-study variability in the IC50 meta-regression, the central estimates of the IC50 have high certainty because of the large number of studies included in the analysis and the random-effects that account for the between study uncertainty. Since the results are so consistent regardless of whether we include or exclude the x-axis uncertainty, we have included the above model with x-axis uncertainty in the supplement as an important validation, but retain the more typical maximum likelihood regression approach for all model fitting throughout the text as this is more consistent with general expectations in the field.

We feel the above analyses have greatly strengthened the research by testing the sensitivity of our analysis on assumptions of the use of the IC50 and the uncertainty in the meta-analysis of the IC50. In addition, we have now discussed other potential limitations with the IC50 analysis and difficulty to calibrate particular assay's in the main text:

Line 333: The analysis presented here has a number of limitations. Firstly, our dose-response analysis requires comparison of the in vivo measured antibody concentrations and the estimated in vitro IC50s. This relied on a meta-regression of estimates of the IC50 of each mAb against different variants, and we observed here a large between-study variation (Figure S1). This means that the results of any particular study or assay may vary quite considerably from the central estimate of the IC50 for analysis of all studies. Despite the limitations of the IC50 meta-analysis, we found that the fitted model for the relationship between antibody concentration and efficacy was robust to the uncertainties in the IC50s (Table S7 and Supplementary Methods). Further, when restricting the analysis to only studies conducted when non-VOC predominated, unadjusted antibody concentrations were similarly predictive of efficacy (Table S3).

3. nAb dynamics is not given per patient. Obviously, this is important to predict how a given individual responds to the nAb therapy (protected or not). Fig 1 data must be improved to include as much as possible of individual Ab dynamics, and quantified using population-based approaches, e.g., mixed effects.

We thank the reviewer for this suggestion. However, individual level data was not supplied upon request by the authors of the original study (as detailed above). Thus, our work only

relates the average concentration with overall efficacy outcomes by time point. We have added a discussion of this limitation:

Line 345: An additional limitation is that we did not have access to raw data from the clinical studies and relied on extraction of data from the published reports. This involved manual extraction of data from figures in some cases, which carries implicit risks of error. Data was extracted independently by two authors, and the results compared to resolve discrepancies⁶. Further, we are reliant upon population level rather than individual level data on antibody concentration, half-lives and clinical outcomes broken down by time intervals as reported in, or extracted from, the published studies. Thus, we could not account for between-subject variability, or subjects lost to follow-up (although fortunately these numbers are relatively small). Additionally, the analysis is strongly influenced by the results from the Herman et al.⁷ and Schmidt et al.⁴ studies, given their contribution of data at lower effective antibody concentrations (Figure 2). It is also evident that each study data point on efficacy contained considerable uncertainty (with wide confidence intervals), this contributed somewhat to wide confidence intervals in the overall fitted model (Figure 2). To gain more precise estimates of the dose-response curve for the use of monoclonal antibodies for prophylaxis, combining the results from more studies, preferably with individual-level data available, and with longer follow-up times (where this is ethical) would be helpful.

4. Only one model for Ab protection is considered (line 391 but equations are not numbered!). Why is this the right model? What about alternative models that may be consistent with the data? For example, a threshold model may be reasonable, or hill-like function (e.g., PMID: 32453765). A note - estimate for k includes 1, so it seems that the model in which $k=1$ should fit the data fine and is a simpler model.

We thank the reviewer for their comments. We have now fitted a range of potential models as suggested by the reviewer (Table S5). We found that all models performed similarly well (AIC differences ≤ 3.5), and thus, given the limited data, we could not distinguish between the potential functional forms. We choose the model with the lowest AIC, which is the same model we used in the original submission (logistic model), but with the maximum efficacy set to 1 (Table S5). We now detail this in the main text and provide the fitted model comparisons (see below also). This change has had the advantage of simplifying the model used in the manuscript (one less parameter), and bringing it in line with the model used for the vaccine efficacy correlate of protection⁸. We feel these changes have greatly strengthened and simplified the work and make it both more rigorous and more accessible to readers.

Efficacy model comparison

Model	Model equation	Parameters	AIC
Logistic model with maximal efficacy 1 (used in the revised manuscript)	$\frac{1}{1 + \exp\left(-k \times (\log_{10}(c) - \log_{10}(c_{50}))\right)}$	k : slope parameter c ₅₀ : concentration at which the efficacy is 50%	124.6
Logistic model (used in the original submission)	$\frac{m}{1 + (2m - 1) \times \exp\left(-k \times (\log_{10}(c) - \log_{10}(c_{50}))\right)}$	m : maximal efficacy k : slope parameter c ₅₀ : concentration at which the efficacy is 50%	126.4
Logistic model with slope 1 (suggested by reviewer #1)	$\frac{m}{1 + (2m - 1) \times \exp\left(-(\log_{10}(c) - \log_{10}(c_{50}))\right)}$	m : maximal efficacy c ₅₀ : concentration at which the efficacy is 50%	125.4
Threshold model constant efficacy below and above the threshold concentration (suggested by reviewer #1)	$\begin{cases} e_{\text{below}}, & \text{if } c < c_{\text{thr}} \\ e_{\text{above}}, & \text{else} \end{cases}$	c _{thr} : threshold concentration e _{below} : efficacy below the threshold e _{above} : efficacy above the threshold	125.2
Double logistic model Logistic model with a change of the slope at the threshold concentration (from the reference suggested by reviewer #1)	$\begin{cases} \frac{m}{m + \exp(-k_1 \times \log_{10}(c))}, & \text{if } c < c_{\text{thr}} \\ \frac{m}{m + \exp(-k_1 \times \log_{10}(c_{\text{thr}}) - k_2 \times (\log_{10}(c) - \log_{10}(c_{\text{thr}})))}, & \text{else} \end{cases}$	m : maximal efficacy k ₁ , k ₂ : slope parameters c _{thr} : threshold concentration	128.1
Exponential model Exponential decay of the relative risk of symptomatic infection with the log ₁₀ -concentration of the monoclonal antibody (from the reference suggested by reviewer #1)	$1 - \exp(-k \times \log_{10}(c))$	k : slope parameter	126.8

Table 1 Comparison of the fit of different efficacy functions to the data. We considered six different models for the protective efficacy of prophylactic mAb treatment depending on the mAb concentration (*c*) and compare them using the AIC. The best fit for each of these models is shown in **Figure 1**.

Figure 1 Different efficacy functions fit to prophylactic mAb treatment data. The different models and their AICs are specified in **Table 1**.

5. How do we know that the model fits the data well? The presented data are very noisy so perhaps the model does fit the data. Providing some statistical evidence of model fit quality is needed - e.g., using chi-square test?

We thank the reviewer for their comment and have now included an assessment of goodness of fit using a chi-squared test.

Line 146: Fitting this logistic dose-response relationship to the data, we estimate the concentration for 50% efficacy of 96-fold the in vitro IC50 (95% CI: 32 – 285) (Table S6, Pearson’s goodness-of-fit test, $\chi_{22} = 19.1$, $p=0.64$, Figure 2).

6. When comparing protection by nAbs and the vaccine authors must investigate i) what sample size would be required to detect a given difference in vaccine vs. nAb efficacy (power analysis)? ii) how do errors in estimating IC50 may contribute to not detecting a difference between two efficacy estimates?

We thank the reviewer for this comment. We note that in comparing these two curves although we did not find a significant difference in some parameters, we agree with the reviewer we had limited power to detect such a difference and this should be appropriately quantified. A sample size calculation for this meta-analysis of studies is prohibitively complex as it requires defining how many studies, and each study may have a different size, different number of time points and follow-up interval. Instead, we followed the reviewer’s advice below and re-parameterised our model to estimate the difference between the parameters for the mAb and vaccine curves and the CIs associated with these differences. The CIs of the differences between the mAb and vaccine curve give an estimate of how different the vaccine and mAb curves could be without us being able to detect the difference. We have now included these estimates of certainty, and a discussion of the extent to which these two relationships could be different and we would not be powered to observe it here in this study.

Line 239: There was no evidence for a difference in the neutralization titer required for 50% protection between vaccination and mAb treatment (fold-change in titre for 50% protection in mAb compared to vaccination is 0.81, 95% CI: 0.26-2.51, Figure S4). However, given the limited power, these results show that if a difference between these groups exist, the fold difference is unlikely to be lower than 0.26 or higher than 2.5. Further, our analysis showed that the best-fit model was one where the same dose-response relationship existed for both vaccination and mAbs but with the estimated slope being higher for vaccination (Figure 4, Table S9).

7. Authors need to perform some type of uncertainty analysis for all major claims to find conditions under which these break. For example, statement that nAbs and vaccine-induced protection are similar - how wrong can you be?

We thank the reviewer for this comment, and as above we have now provided measures of certainty for all major claims. When this is significant association or correlation we provide statistical tests, p-values etc, when we observe an absence of a detectable difference we now report the CIs as an estimate of how extreme the difference could be without us being likely to detect that difference.

Minor concerns

All equations must be numbered and referred to in the text by numbers.

We have now numbered all equations in the text.

What happens if you do not normalize the data by IC50? Does the analysis fail?

We thank the reviewer for the suggestion. As above, when we do not normalise by the IC50 the correlation between antibody concentration and protection remains significant if focusing the analysis only on studies looking at non Variants of Concern, and we now discuss this in the main text (as in response to point 2 above).

It is unclear in how graphs in Fig 3 are useful. Are we concerned with a particular level of efficacy? Should one actually show how often Abs need to be injected to maintain a level of efficacy? Also, scales for x axes in 1-c should be the same.

The purpose of these figures is to demonstrate how the model can be used to predict whether an antibody will continue to be protective against a new variant and, if so, for how long that efficacy will last (above a specified threshold, which we choose as 50% for the purpose of an illustration here). A different threshold could equally be used, and we now make this clear in the main text. Figure 3d indicates how frequently treatment would need to be repeated in order to maintain efficacy above our specified target of 50%. We have clarified how one might use the observations in figure 3 or a similar analysis in the main text. We have now aligned the x-axes of figure 3 (a-c) to be uniform.

Line 172: For example, cilgavimab/tixagevimab administered intramuscularly at a dose of 300 mg is predicted to maintain >50% protection for 581 days (95% CI: 433 – 730 days) against the ancestral variant, since the in vitro IC50 is 4.27 ng/mL and the half-life of this antibody combination is 95 days (Figure 3, Tables S8, Figure S3). However, given the in vitro IC50 increases 8.9-fold to Omicron BA.2, it is predicted that this same dose would only provide protection above 50% for 282 days (95% CI: 133 – 430 days) against this variant. In this example, this mAb combination would need to be administered every 282 days in order to maintain at least 50% efficacy against Omicron BA.2. Importantly, cilgavimab/tixagevimab is not predicted to attain 50% efficacy against Omicron BA.1 even shortly after treatment (because of the large increase in the in vitro IC50 to this variant (Supplementary Table S3)). Similar estimates can be determined from this analysis for the other mAbs based on the in vitro IC50 of these mAbs to different SARS-CoV-2 variants (Figure 3).

Another formulation of this question is to ask “What is the maximum increase of IC50 (drop in neutralisation titre) that can be tolerated while still maintaining a minimum duration of protection?”. For example, if we wish to provide a period of at least 30 days with >50% protection, then cilgavimab/tixagevimab, casirivimab/imdevimab and adintrevimab, at the current doses, can tolerate at most 56.5-fold (95% CI: 19.1 – 167.4), 143.8-fold (95% CI: 48.5 – 426.2) and 61.1-fold (95% CI: 20.6 – 181.3) increases in in vitro IC50 compared to the in vitro IC50 against the ancestral variant, respectively. Figure 3d shows the predicted duration of >50% protection for casirivimab/imdevimab, cilgavimab/tixagevimab, and adintrevimab for any given fold-change in in vitro IC50. Using this analysis, we see that all of these mAb are predicted to be ineffective against at least some of the recent circulating Omicron subvariants (e.g. BA.2.75, BQ.1.1 and XBB, where data on IC50 shift is available),

because of larger shifts in the IC50 (Table S3). Going forward, this analysis will provide drug developers with a means of using in vitro neutralization data to predict the efficacy of candidate broadly neutralizing mAb against novel SARS-CoV-2 variants, as well as to guide dosing/dosing interval decisions for promising monoclonal antibodies in order to achieve a specified level of protection against the most relevant circulating variants.

line 136 - half-life of this Ab seems longer than the typical 2-3wks. Why?

We thank the reviewer for this comment, two of the three antibody products were developed especially with modified Fc-receptors to increase the half-life. We now explain this in the text.

Line 320: In particular, adintrevimab, tixagevimab, and cilgavimab all have modified Fc receptors to increase the antibody half-life^{4, 9}, and this likely restricts some Fc related antibody functions.

Line 237-240: I think this will confuse a lot of people. I would suggest more about practical aspects - e.g., if we want to protect a person from SARS-CoV2, we would need to inject a given Ab at dose X every day Y to maintain Z level of protection. That people will understand.

We thank the reviewer for this suggestion. We have updated this section to improve clarity.

Line 172: For example, cilgavimab/tixagevimab administered intramuscularly at a dose of 300 mg is predicted to maintain >50% protection for 581 days (95% CI: 433 – 730 days) against the ancestral variant, since the in vitro IC50 is 4.27 ng/mL and the half-life of this antibody combination is 95 days (Figure 3, Tables S8, Figure S3). However, given the in vitro IC50 increases 8.9-fold to Omicron BA.2, it is predicted that this same dose would only provide protection above 50% for 282 days (95% CI: 133 – 430 days) against this variant. In this example, this mAb combination would need to be administered every 282 days in order to maintain at least 50% efficacy against Omicron BA.2. Importantly, cilgavimab/tixagevimab is not predicted to attain 50% efficacy against Omicron BA.1 even shortly after treatment (because of the large increase in the in vitro IC50 to this variant (Supplementary Table S3)). Similar estimates can be determined from this analysis for the other mAbs based on the in vitro IC50 of these mAbs to different SARS-CoV-2 variants (Figure 3).

It is nice that authors list limitations. But there are more of these – see comments above about the data in particular.

We thank the reviewer for highlighting additional limitations for the discussion. We have now expanded this discussion, especially with regard to the raw data being unavailable and thus extracted.

Line 345: An additional limitation is that we did not have access to raw data from the clinical studies and relied on extraction of data from the published reports. This involved manual extraction of data from figures in some cases, which carries implicit risks of error. Data was extracted independently by two authors, and the results compared to resolve discrepancies⁶. Further, we are reliant upon population level rather than individual level data on antibody concentration, half-lives and clinical outcomes broken down by time intervals as reported in, or extracted from, the published studies. Thus, we could not account for between-subject variability, or subjects lost to follow-up (although fortunately these numbers are relatively small). Additionally, the analysis is strongly influenced by the results from the Herman et al.⁷ and Schmidt et al.⁴ studies, given their contribution of data at lower effective antibody concentrations (Figure 2). It is also evident that each study data point on efficacy contained considerable uncertainty (with wide confidence intervals), this contributed somewhat to wide confidence intervals in the overall fitted model (Figure 2). To gain more precise estimates of the dose-response curve for the use of monoclonal antibodies for prophylaxis, combining the results from more studies, preferably with individual-level data available, and with longer follow-up times (where this is ethical) would be helpful.

Line 326 - 328 - authors provide no evidence that this approach mitigates systemic biases in the analysis.

We have amended this discussion in keeping with the other comments on robustness of the results to the meta-regression of the IC50 values.

Line 333: The analysis presented here has a number of limitations. Firstly, our dose-response analysis requires comparison of the in vivo measured antibody concentrations and the estimated in vitro IC50s. This relied on a meta-regression of estimates of the IC50 of each mAb against different variants, and we observed here a large between-study variation (Figure S1). This means that the results of any particular study or assay may vary quite considerably from the central estimate of the IC50 for analysis of all studies. Despite the limitations of the IC50 meta-analysis, we found that the fitted model for the relationship between antibody concentration and efficacy was robust to the uncertainties in the IC50s (Table S7 and Supplementary Methods). Further, when restricting the analysis to only studies conducted when non-VOC predominated, unadjusted antibody concentrations were similarly predictive of efficacy (Table S3).

Figure 1 must have same scales for all panels for better interpretation.

The scales of all panels in Figure 1 are now the same.

Line 112 (supplement): The 95% CIs for efficacy are likely to be correct only approximately. Perhaps using resampling/bootstrap is a better approach.

We thank the reviewer for this comment. We note that the 95% CI calculation referred to here are only used for visualising the CIs of the efficacy data in figure 2 and 4. When fitting the data, the raw count data is used. Thus, we feel that the approximation used here is adequate for the purposes of visualising the uncertainty.

Table S3 - leave one out. What about leaving 2 out? 3 out? This will verify robustness of the conclusions.

We have now performed a leave-N-out analysis considering all cases with as few as two studies. As expected, we find that the two studies (Schmidt et al. and Herman et al.), which provide evidence of lower efficacy, are critical for detecting a significant correlation between antibody conc. and efficacy. If both of these studies are missing, we see no significant correlation, but in all other leave-N-out cases we find a significant correlation (Table S4). This is now discussed in the discussion:

Line 353: Additionally, the analysis is strongly influenced by the results from the Herman et al.⁷ and Schmidt et al.⁴ studies, given their contribution of data at lower effective antibody concentrations (Figure 2).

Table S4 - efficacy cannot be more than 100%!

We have now amended our model fitting procedure to prevent efficacies >100% in all fitted models (Table S5).

Table S7 – how relevant is peak nAb concentration? Seems that the whole kinetics should be more important than the peak. Please investigate.

As suggested, we have removed the discussion focused on the peak nAb concentration and instead focus on time of efficacy above a target threshold of 50% in the results.

Table S6 - numbers look weird, references seem like a "power".

We have now moved the literature references to a separate row to clearly show they are references and not powers.

Reviewer #2 (Remarks to the Author):

Thank you for inviting me to review this manuscript. This work follows on from previous analyses by the same authors on the dose-response relationship between MAb dose after conversion to neutralizing dose equivalence, and protection from hospitalisation when MAb given early after symptomatic SARS-CoV-2 infection. The use of the dose-response relationship between normalized IC50 and efficacy to predict the clinical efficacy of a new variant against existing MAbs is particularly important and informative. The authors aggregate data from 5 MAb RCTs with data on clinical efficacy alongside PK data, to a range of MAbs.

This manuscript has considerable practical implications for how existing and future MABs are authorised/continued to be used in treatment moving forwards.

We thank the reviewer for their supportive comments.

I only have minor comments and suggestions:

1. Please state whether every participant in each of the studies had samples taken for PK data or just a subset.

We thank the reviewer for this comment, and have now added more detail on each study in the table of studies (Table S2), including an indication of which studies contained a subset of samples or every individual. Note that we only used the mean of reported antibody concentration at each time point and match this to the overall observed efficacy by time interval, thus we are not matching individual's antibody concentrations with their outcomes since this data was not available.

2. Figure 1 - add 'every 4 weeks' to 1c 'repeated administration' so clear by just looking at figure when the doses would be

We appreciate this suggestion and have updated the figure accordingly.

3. For generation of data in figure 2, state in main text that the timeframes/windows of efficacy and antibody concentration for each study used are found in Table S1

We agree and have now made this clear in the main text:

Line 104: To investigate whether declining efficacy with time and new variants was indicative of a dose-response relationship between mAb concentration and efficacy, we compared the antibody concentrations reported within different time intervals in each study with the reported efficacy at the corresponding time interval (details of time intervals used in analysis provided in Table S1).

4. Data on antibody efficacy vs IC50 - it isn't clear in the results or figure 2 legend whether the X axis data (i.e. fold in vitro IC50) were generated using normalised IC50 to multiple variants or just to ancestral virus. To include Schmidt et al data (where efficacy against Delta and Omicron are what is available) I assume it may be the former. If IC50 fold against multiple variants was amalgamated, how did the authors deal with the issue of IC50 above the limit of the in vitro assay used in the case of multiple omicron-MAB combinations? What would make most sense is to only include IC50 fold data for viruses that were circulating at the time any particular study was conducted (so matched to the efficacy data). This may be what the authors did but it needs clarifying.

The reviewer is correct that the analysis we performed previously did take variant IC50 into account. I.e. we matched the strains circulating during the RCT efficacy outcomes with the variants used in assessing the IC50 for each antibody. This was with the exception that in the

previous IC50 meta-analysis we did not have data on neutralisation against delta and so we had assumed neutralisation IC50 for delta was the same as WT. However, we have now improved this analysis by including a revised and more comprehensive meta-analysis of IC50 from the literature using the Stanford database (<https://covdb.stanford.edu/>)⁵. Thus, our updated meta-regression now allows us to match the predominant circulating variant from each RCT with the appropriate variant specific IC50. The matched IC50 estimate with the circulating variant for each study is explained in the results and detailed in Table S2.

Line 119: To adjust for the different neutralising potencies of each antibody and loss of potency against different variants, we normalized antibody concentration using the in vitro IC50 for each antibody against the dominant variant circulating at the time of the study (Table S2). These in vitro IC50 for different antibody / variant combinations were obtained from a meta-analysis of in vitro studies (Table S3 and Figure S1, using data from the Stanford University Coronavirus and Resistance Database⁵). We found that after normalising by the in vitro IC50, we found a significant relationship between efficacy and mAb concentration (as a fold of the in vitro IC50) (RR=0.40 per log₁₀-increase in antibody concentration, p<0.0001). Together this suggests that in vivo monoclonal antibody concentrations adjusted by neutralising potency is correlated with efficacy to prevent COVID-19.

5. Some discussion of how some design aspects of the primary studies may have affected the estimates would be useful:

a. To what extent did they recruit seronegative participants only (and therefore the efficacy can be attributed solely to the MAb)

b. To what extent did they allow vaccination to occur or include vaccinated participants? This is important in considering the efficacy retained in the face of lower MAb antibody concentrations, and introduce greater uncertainty to the estimates

We thank the reviewer for this comment, and have now added this information for each study to the table S2, and discussed differences in study design in the main text. Of note, there is one study where vaccination was allowed in follow up. However, we note that this has likely had minimal impact on the efficacy estimates (Table S2). Additionally, we note that the correlation between efficacy and antibody concentration was robust to exclusion of any one study (Table S4, leave-one-out analysis). We now discuss this in the discussion:

Line 363: Other differences between the study designs used in our analysis may have impacted the analysis, which we could not account for directly. For example, seropositivity was zero at baseline in all studies except Isa et al.¹⁰, and vaccination of participants after treatment was allowed in Herman et al.⁷ (Table S2). Fortunately, in our sensitivity analysis we found that the correlation between antibody correlation and efficacy was robust to removing either of these studies (Table S4). Also, uptake of vaccination in Herman et al. was similar between treated and control arms (~35%). This likely lowered the power of the study to detect efficacy at later time points, but otherwise was not expected to have a large impact on the efficacy estimates in this study (Table S2). In addition, studies used different modes of delivery of the

monoclonal antibodies, i.e. casirivimab/imdevimab was administered subcutaneously in the Isa¹⁰, O'Brien¹¹ and Herman⁷ studies, whereas cilgavimab/tixagevimab and adintrevimab were administered intramuscularly^{12,4}. Plasma antibody concentrations appear to increase more slowly following intramuscular administration compared with subcutaneous administration of casirivimab/imdevimab (Figure 1 and Table S8), and thus it is possible that there is a delay until protective antibody concentrations are achieved. To avoid this difference and also to account for the risk of infection around the time of antibody administration, we omitted the earliest time interval from our analysis (which encompassed the first 7, 10, 27, 28, 30, or 90 days across different studies, Table S1).

6. In previous publications, the authors have elegantly used a fold-convalescent approach to normalise data from several studies and estimate the relationship between in vitro Nab titre and clinical efficacy. The same approach is used here to compare mAb and vaccine efficacy and I appreciate the value of doing so. Do the authors have data that would allow them to go further and standardise their output (at least for some MAb against ancestral virus) to WHO standard 20/136 IU50/mL by estimating the relationship between antibody concentration and neutralisation titre? Or the newer 21/338 (variant) standards? It would help with comparison with other studies reporting on correlates of protection in IU50/mL.

We appreciate the reviewer's suggestion. However, unfortunately, we are unable to provide such an analysis, since the Stanford database of IC50 measurements did not contain neutralisation data that had been normalised by the WHO standards. Interestingly, the analysis of the (notionally) identical monoclonal antibody products across multiple assays and labs can act as a standard in itself. Thus, one way to interpret the meta-regression of IC50 data reported here, is that the antibodies provide an assessment of the degree of standardisation of assays. However, testing whether this approach of using IC50 of a panel of mAb to normalise between assays is comparable to using a WHO standard is not something we are able to test here, thus we have retained the fold-convalescent scale.

Reviewer #3 (Remarks to the Author):

In this manuscript, Stadler and colleagues extract data on protection from symptomatic SARS-CoV-2 infection and antibody pharmacokinetics from clinical trials of monoclonal neutralizing antibodies used for pre- and peri-exposure prophylaxis. Based on this data, they model a relationship between antibody concentrations relative to in vitro IC50s and estimated protective efficacy, including longitudinally for individual antibodies/antibody combinations. Finally, they model antibody efficacy for neutralizing activity relative to neutralization seen in convalescent individuals and compare this data to similar data previously determined for vaccinees.

The analysis is overall valuable because it attempts to identify mAb-mediated serum neutralization titers that provide protection from symptomatic COVID-19 when administered as prophylaxis. Particularly in individuals with insufficient responses to vaccination, the results can support the development of strategies for antibody prophylaxis

(e.g., dosing, frequency of dosing, adjustments for newly emerging variants).

We thank the reviewer for their positive comments.

Limitations and suggestions:

- Because of the overall limited number of cases, the efficacy confidence intervals for individual time periods are very wide (often ranging from almost 0% to 100% for the peer-reviewed data) (Figure 1; Figure 4). This applies as well to the overall predicted IC50 for 50% protection (>1 log range in the CI). This imprecision could be emphasized somewhat more in the limitations section.

We thank the reviewer for these comments. Given the comments from Reviewers 1 and 3, we have now emphasised this imprecision and the source of these imprecisions in more detail in the discussion:

Line 345: An additional limitation is that we did not have access to raw data from the clinical studies and relied on extraction of data from the published reports. This involved manual extraction of data from figures in some cases, which carries implicit risks of error. Data was extracted independently by two authors, and the results compared to resolve discrepancies⁶. Further, we are reliant upon population level rather than individual level data on antibody concentration, half-lives and clinical outcomes broken down by time intervals as reported in, or extracted from, the published studies. Thus, we could not account for between-subject variability, or subjects lost to follow-up (although fortunately these numbers are relatively small). Additionally, the analysis is strongly influenced by the results from the Herman et al.⁷ and Schmidt et al.⁴ studies, given their contribution of data at lower effective antibody concentrations (Figure 2). It is also evident that each study data point on efficacy contained considerable uncertainty (with wide confidence intervals), this contributed somewhat to wide confidence intervals in the overall fitted model (Figure 2). To gain more precise estimates of the dose-response curve for the use of monoclonal antibodies for prophylaxis, combining the results from more studies, preferably with individual-level data available, and with longer follow-up times (where this is ethical) would be helpful.

- Many cases/breakthrough infections are derived from the Schmidt et al. study on adintrevimab, which is – as of yet – not published in peer-reviewed form (i.e., published data may change).

We thank the reviewer for this comment. Fortunately, the study is now published in *Science Translational Medicine* and we have re-extracted from this published version of the manuscript, and note that the numbers have not changed since our previous submission.

- Data on the casirivimab/imdevimab combination were collected both in a pre- and in a peri-exposure setting, whereas for the other antibodies, only the pre-exposure setting was investigated. Because these are different clinical scenarios, they cannot necessarily be

compared. It seems more prudent to only analyze the pre-exposure setting.

We thank the reviewer for suggesting this potential confounder and sensitivity analysis. We would like to note that even in cases with peri-exposure use of the mAb, we intentionally exclude all case data that occurred in at least the first week, since this would likely be the result of infection events that may have occurred prior to receiving the mAb, whereas later infections were very unlikely to have been the result of an infection occurring prior to mAb treatment. However, we agree that the peri-exposure setting is still different from true prophylaxis and that we may not be able to account for that difference perfectly. Thus, we now show that the correlation between efficacy and antibody concentration is still significant when we only analysis studies of true prophylaxis:

Line 130: To test the robustness of this correlation we performed sensitivity analyses. Firstly, our analysis uses a combination of data from true pre-exposure prophylaxis settings and also from settings of peri-exposure prophylaxis (i.e. in individuals after some degree of known contact with a COVID-19 index case). Thus, we repeated the analysis when only true pre-exposure prophylaxis studies were included and found the relationship remained significant (RR=0.45 per log₁₀-increase in antibody concentration, p<0.0001). Further, in a leave-one-out, leave-two-out and so on analyses, we found this relationship remained significant in all cases except when both the Schmidt et al.⁴ and Herman et al.⁷ studies were omitted (Table S4). Suggesting a sensitivity of the results to these two studies.

- In the Herman et al. study of prolonged follow-up after a single casirivimab/imdevimab injection, participants were eligible for COVID-19 vaccination, and 35% in the antibody group received at least one vaccination during follow-up (similar number in the placebo group) and there were almost no more symptomatic infections after vaccination. Did the authors account for this?

We agree this is a limitation of our analysis which had not been discussed. We now explain this limitation. We note, that because so few events (3) occurred in vaccinated individuals and only ~35% of participants were vaccinated (and in nearly equal frequencies in control and treatment arms) this is unlikely to have a significant impact on the overall efficacy estimates (other than reducing power of the study because of lower risk in the vaccinated group) (Table S2). We now also discuss this limitation in the discussion:

Line 363: Other differences between the study designs used in our analysis may have impacted the analysis, which we could not account for directly . For example, seropositivity was zero at baseline in all studies except Isa et al.¹⁰, and vaccination of participants after treatment was allowed in Herman et al.⁷ (Table S2). Fortunately, in our sensitivity analysis we found that the correlation between antibody correlation and efficacy was robust to removing either of these studies (Table S4). Also, uptake of vaccination in Herman et al. was similar between treated and control arms (~35%). This likely lowered the power of the study to detect efficacy at later time points, but otherwise was not expected to have a large impact on the efficacy estimates in this study (Table S2).

- Efficacies of mAb administration and vaccination are compared by analyzing neutralization titers relative to those seen in convalescent individuals. For the vaccinees, previously published data from Khoury et al, Nat Med 2021, were used. This data was based on studies in which vaccinated individuals and convalescent individuals were tested side-by-side (all live virus). In contrast, for the assessment of mAb efficacy as related to fold-convalescent neutralization, the data for convalescent neutralization were derived from a wider range of neutralization studies (refs 7-17 in supplement). These studies were not related to the monoclonal antibody infusion trials and used more diverse assays (pseudovirus vs. live virus; focus reduction vs. luminescence; etc.) Neutralization assays can differ in the effects of individual antibody contributions (e.g., pronounced activity for RBD-targeting neutralizing antibodies – such as the monoclonal antibodies tested here – in assays using ACE2-overexpressing cells). The authors discuss assay differences in the context of mAbs (lines 318 ff.). However, they should also comment on how the comparison between vaccinated and mAb-treated individuals in Figure 4 might be affected (curve for vaccinated is based on comparison of polyclonal vs. polyclonal antibodies; curve for antibody group based on neutralization assays using RBD-targeting antibodies vs. polyclonal antibodies).

We thank the reviewer for noting this limitation that was not previously discussed. However, we would like to make a clarifying point, that the convalescent plasma assessments here were always paired with the mAb assessments, just as the vaccine and convalescent plasma samples were matched in the previously Khoury et al. study. As noted by the reviewer, a difference here is that there are many studies of IC50 included for each monoclonal antibody rather than only a single paper. This is because at the beginning of the pandemic there was very little data on antibody responses to vaccination, whereas now there are many estimates of mAb IC50. Thus, we have been as comprehensive as possible in our meta-analysis of IC50, while still matching convalescent plasma titers with mAb IC50 estimates reported in the same paper. We note that not all papers had convalescent plasma panels but our mixed effects modelling approach accounts for the repeated measures across any one study, and provides a robust approach to obtain a central estimate of the IC50 in this meta-analysis without biasing the result by the lack of independence between repeated measures in the same study. We now discuss these differences and similarities:

Line 382: Finally, when comparing mAb and vaccination, there are some additional limitations. In particular, this comparison involves converting the antibody concentration data to a fold of convalescent scale. However, it is possible that the normalization employed here oversimplifies the comparison since it does not appreciate potential differences between neutralization readouts for polyclonal sera and monoclonal antibodies. Further, only a subset of studies (n=19) in our meta-regression reported the geometric mean neutralization titer of suitable panel of convalescent sera. In addition, the definition of convalescent sera was specified differently in each study, introducing some potential confounders to these aggregated estimates. Even so, this approach has revealed surprisingly similar prophylactic and vaccine efficacy for a given neutralization titer on the fold of convalescent scale.

- In Figure 4B, very little information below 0.1-fold-convalescent neutralization is available (none at all for the vaccinees). In the original Khoury et al., Nat Med 2021, paper, the curve for vaccinees is cut-off at 0.1. It might be prudent to do so here as well (show curves up to the lowest available data point).

We thank the reviewer for this suggestion and have now truncated the graph further to the right on the x-axis.

More minor:

- Figures 4B and S3 appear to be identical.

We have removed figure S3.

- In Figure S1, serum neutralization titers from convalescent individuals (in blue) are shown with the same y-axis as monoclonal antibodies. However, this data is probably ID50s (dilutions) rather than concentrations. This should be corrected/not shown with the same y-axis.

We apologise for the confusion – we intended for the “blue” axis label to correspond to the convalescent plasma (titre) and black for the IC50 of mAb. We have now amended this to improve clarity.

- In addition to neutralization, antibodies may act through Fc-mediated effector function. Compared to casirivimab/imdevimab and adintrevimab, tixagevimab and cilgavimab have been Fc-modified for reduced Fc interactions. In contrast, half-life extending mutations have been introduced into adintrevimab as well as tixagevimab and cilgavimab; these mutations may also affect FcRn-mediated mucosal deposition. These differences may complicate the comparison of different antibodies and could be briefly discussed.

We have now discussed these differences in more detail and highlighted these as potential reasons for the lower efficacy of mAb at high neutralisation titres compared to vaccination.

Line 317: The difference in protection at a given neutralization titer between vaccination and monoclonal antibody therapy may be due to the additional benefit in vaccinees of a polyclonal antibody response, other non-neutralizing functions of antibodies, recall of immune memory, and/or other cellular immune responses. In particular, adintrevimab, tixagevimab, and cilgavimab all have modified Fc receptors to increase the antibody half-life^{4,9}, and this likely restricts some Fc related antibody functions. These functions may contribute to the estimated trend towards higher protection for vaccines at high neutralizing titers. While our analysis has shown that neutralizing antibodies alone are sufficient to provide a high level of protection from COVID-19 at the neutralisation titres induced by vaccination, it is not possible to conclude from this analysis that neutralizing antibodies are necessary for protection.

Also, since casirivimab and imdevimab have intact Fc-receptors, differences in efficacy between these products may arise from other non-neutralizing functions that we have not directly considered here. We note that evidence in animal models supports the findings that neutralizing antibodies mediate protective immunity¹³, with some showing an additional benefit of Fc-receptor function¹⁴.

References

1. Rubin D. Chapter 2: Multiple imputation. In: *Flexible Imputation of Missing Data (Second Edition)* (Edited by: van Buuren S). Chapman and Hall (2018).
2. Khoury DS, et al. Correlates of Protection, Thresholds of Protection, and Immunobridging among Persons with SARS-CoV-2 Infection. *Emerg Infect Dis* **29**, 381-388 (2023).
3. Organization WH. Establishment of the WHO International Standard and Reference Panel for anti-SARS-CoV-2 antibody. *Expert Committee on Biological Standardization, Geneva*, 9-10 (2020).
4. Schmidt P, et al. Antibody-mediated protection against symptomatic COVID-19 can be achieved at low serum neutralizing titers. *Sci Transl Med* **15**, eadg2783 (2023).
5. Tzou PL, Tao K, Pond SLK, Shafer RW. Coronavirus Resistance Database (CoV-RDB): SARS-CoV-2 susceptibility to monoclonal antibodies, convalescent plasma, and plasma from vaccinated persons. *PLoS One* **17**, e0261045 (2022).
6. Li T, Higgins, JPT, Deeks, JJ. Chapter 5: Collecting data. In: *Cochrane Handbook for Systematic Reviews of Interventions (Version 6.3)* (Edited by: Higgins J, Thomas, J) (2022).
7. Herman GA, et al. Efficacy and safety of a single dose of casirivimab and imdevimab for the prevention of COVID-19 over an 8-month period: a randomised, double-blind, placebo-controlled trial. *Lancet Infect Dis* **22**, 1444-1454 (2022).
8. Khoury DS, et al. Neutralizing antibody levels are highly predictive of immune protection from symptomatic SARS-CoV-2 infection. *Nat Med* **27**, 1205-1211 (2021).
9. Loo YM, et al. The SARS-CoV-2 monoclonal antibody combination, AZD7442, is protective in nonhuman primates and has an extended half-life in humans. *Sci Transl Med* **14**, eab18124 (2022).
10. Isa F, et al. Repeat subcutaneous administration of casirivimab and imdevimab in adults is well-tolerated and prevents the occurrence of COVID-19. *Int J Infect Dis* **122**, 585-592 (2022).

11. O'Brien MP, *et al.* Subcutaneous REGEN-COV Antibody Combination to Prevent Covid-19. *N Engl J Med* **385**, 1184-1195 (2021).
12. Levin MJ, *et al.* Intramuscular AZD7442 (Tixagevimab-Cilgavimab) for Prevention of Covid-19. *N Engl J Med* **386**, 2188-2200 (2022).
13. McMahan K, *et al.* Correlates of protection against SARS-CoV-2 in rhesus macaques. *Nature* **590**, 630-634 (2021).
14. Schafer A, *et al.* Antibody potency, effector function, and combinations in protection and therapy for SARS-CoV-2 infection in vivo. *J Exp Med* **218**, (2021).

Reviewers' Comments:

Reviewer #1:
Remarks to the Author:

Positive feedback.

Thank you for taking the comments seriously and addressing some uncertainties with the analysis. I hope that you have a positive view of this review process (I typically don't because often reviewers don't get the meaning of our work.)

Additional major concerns

Authors clearly softened the message in the text about similarity of nAb and vaccine-mediated protection may be due to low statistical power. However, this is not stated clearly in the abstract. The point that no difference between nAb and vaccine mediated protection could be due to low power has to be clearly stated. Specifically, you could say something like this: "We find no evidence for a difference between the 50% protective titer for monoclonal antibodies and vaccination although this may be due to limited power to detect a relatively small (20% difference) in protection efficacy".

I wonder if using fold-from-convanlesence is part of the issue with having close estimates of efficacy of nAbs and vaccine-induced protection. How do we know this is the right metric? From point of view of rigorous research, the best way is to have individuals randomized to either vaccinated or with injected nAbs, measure nAbs in both cohorts using the same method, and then follow them up to measure protection. Then we will know. I think the issue with "normalization" of Ab titers between absolutely different types of studies must be clearly acknowledged, so the conclusion of similar efficacy can be only made tentatively.

Minor concerns.

I appreciate description of the process of how authors tried to get original data and sympathize with their failures. I think this text MUST be included in the paper as a part of Discussion/limitations. I wonder if citing specific journals where studies have been published could be a good indication that these journals (NEJM, Sci Trans Med, Lancet Inf Dis) are NOT following FAIR principles and should be noted as such publicly. It is unacceptable that in the 21st century it is hard to impossible to get experimental data from published papers.

Line 66 - "fewer trials". Not a good argument. Fewer is better than none.

Line 390-91 - lower stat power could be to blame here.

Figure 1 - all panels should have same range for x axis for better visual comparisons

Figure 2 - put estimated parameters here, either in the figure itself (best) or in caption. It is 1 parameter, so don't ask the reader to search for these in tables/text/etc.

Figure 4a - put fold difference, in addition to the p value (e.g., as $x1.2$ ($p=xxx$)). Fig 4b - using different styles of model fit will help folks with color blindness to tell difference.

Supplement, page 6 - why's is W larger than B (put specific numbers!)? Is there an intuitive explanation of this?

Table S5 - please list best fit parameters here. Also, make a column of delta AIC, so model comparison is made easier.

Figure S2 and Table S5 - I found it interesting that the threshold model I suggested in my review fits the data nearly with identical quality as the main model (it is second best). I think this should be discussed some more - e.g., that you cannot tell which model is the right one. Why do you think threshold model fits the data so well? Adding AIC or delta values to each panel in Fig S2 would be useful, perhaps along with estimated model parameters. You have space in top left corners.

Fig S5 - put AIC values (or better delta AIC) for each panel/model fit.

Reviewer #2:

Remarks to the Author:

The authors have responded to all my comments or provided satisfactory rebuttals as to why this is not possible. The revised manuscript taking into account all reviewers' comments is much improved. I look forward to seeing this manuscript in print.

Reviewer #3:

Remarks to the Author:

The authors have adequately addressed my comments.

A small number of remaining notes:

1. Figure 4B appears to remain unchanged, whereas the rebuttal letter states that it has been truncated "further to the right on the x-axis".

2. The section dealing with potential differences in antibody-mediated Fc effector functions needs some correction.

What has been modified on some of the antibodies are the Fc domains, not the Fc receptors (these are expressed on effector cells).

In addition, while adintrevimab, tixagevimab, and cilgavimab do indeed have Fc domain mutations that increase half-life through enhanced FcRn binding, it is rather unlikely that these mutations will interfere much with Fc effector functions. For adintrevimab, Fc effector functions should be mostly unaffected compared to a regular IgG antibody ("LA" mutation; see also data in the Schmidt et al. paper). For tixagevimab/cilgavimab, mutations that reduced Fc effector functions were specifically introduced ("TM" mutations) and are distinct from the mutations that enhance half-life ("YTE" mutations).

3. Lines 284/285: Instead of writing "approx. 95-fold", I would suggest to write "approx. 100-fold".

REVIEWERS' COMMENTS

Reviewer #1 (Remarks to the Author):

Positive feedback.

Thank you for taking the comments seriously and addressing some uncertainties with the analysis. I hope that you have a positive view of this review process (I typically don't because often reviewers don't get the meaning of our work.)

Additional major concerns

Authors clearly softened the message in the text about similarity of nAb and vaccine-mediated protection may be due to low statistical power. However, this is not stated clearly in the abstract. The point that no difference between nAb and vaccine mediated protection could be due to low power has to be clearly stated. Specifically, you could say something like this: "We find no evidence for a difference between the 50% protective titer for monoclonal antibodies and vaccination although this may be due to limited power to detect a relatively small (20% difference) in protection efficacy".

We thank the reviewer for all their constructive comments. We have now added a statement in the abstract to soften this point. We did not include the "(20% difference) in protection efficacy" comment, since the comment relates to IC-50 levels and it seems unusual to report potential differences in efficacy levels.

Abstract: Multiple monoclonal antibodies have been shown to be effective for both prophylaxis and therapy for SARS-CoV-2 infection. Here we aggregate data from randomized controlled trials assessing the use of monoclonal antibodies (mAb) in preventing symptomatic SARS-CoV-2 infection. We use data on the in vivo concentration of mAb and the associated protection from COVID-19 over time to model the dose-response relationship of mAb for prophylaxis. We estimate that 50% protection from COVID-19 is achieved with a mAb concentration of 96-fold of the in vitro IC50 (95% CI: 32 – 285). This relationship provides a tool for predicting the prophylactic efficacy of new mAb and against SARS-CoV-2 variants. Finally, we compare the relationship between neutralization titer and protection from COVID-19 after either mAb treatment or vaccination. We find no significant difference between the 50% protective titer for mAb and vaccination, although sample sizes limited the power to detect a difference.

I wonder if using fold-from-convalescence is part of the issue with having close estimates of efficacy of nAbs and vaccine-induced protection. How do we know this is the right metric? From point of view of rigorous research, the best way is to have individuals randomized to either vaccinated or with injected nAbs, measure nAbs in both cohorts using the same method, and then follow them up to measure protection. Then we will know. I think the issue with "normalization" of Ab titers between absolutely different types of studies must

be clearly acknowledged, so the conclusion of similar efficacy can be only made tentatively.

We have further included an explicit statement of the problem of normalization in the discussion:

Line 385: Finally, when comparing mAb and vaccination, there are some additional limitations. In particular, this comparison involves converting the antibody concentration data to a fold of convalescent scale. However, we have not shown that such normalization can account for all the differences between neutralization titers achieved by vaccination and mAb administration. For example, it is possible that the normalization employed here oversimplifies the comparison since it does not appreciate potential differences between neutralization readouts for polyclonal sera and monoclonal antibodies. Further, only a subset of studies (n=19) in our meta-regression reported the geometric mean neutralization titer of suitable panel of convalescent sera. In addition, the definition of convalescent sera was specified differently in each study, introducing some potential confounders to these aggregated estimates. Even so, this approach has revealed surprisingly similar prophylactic and vaccine efficacy for a given neutralization titer on the fold of convalescent scale (within the statistical power of the data).

Minor concerns.

I appreciate description of the process of how authors tried to get original data and sympathize with their failures. I think this text MUST be included in the paper as a part of Discussion/limitations. I wonder if citing specific journals where studies have been published could be a good indication that these journals (NEJM, Sci Trans Med, Lancet Inf Dis) are NOT following FAIR principles and should be noted as such publicly. It is unacceptable that in the 21st century it is hard to impossible to get experimental data from published papers.

We have now acknowledged that data was requested and not received:

Line 347: An additional limitation is that we did not have access to raw data from the clinical studies. Data was requested from all corresponding authors of the original studies (31 March 2023) but was not provided by the time of revision (30 June 2023), thus we relied on extraction of data from the published reports. This involved manual extraction of data from figures in some cases, which carries implicit risks of error. Data was extracted independently by two authors, and the results compared to resolve discrepancies³⁸.

Line 66 - "fewer trials". Not a good argument. Fewer is better than none.

We have removed this sentence.

Line 390-91 - lower stat power could be to blame here.

We have included the caveat of statistical power here:

Line 395: Even so, this approach has revealed surprisingly similar prophylactic and vaccine efficacy for a given neutralization titer on the fold of convalescent scale (within the statistical power of the data).

Figure 1 - all panels should have same range for x axis for better visual comparisons

Amended.

Figure 2 - put estimated parameters here, either in the figure itself (best) or in caption. It is 1 parameter, so don't ask the reader to search for these in tables/text/etc.

We added the two estimated parameters of the dose response curve to the caption:

Line 729: The best fit parameters of the dose-response relationship are a concentration for 50% protection of 96.2-fold in vitro IC50 (95% CI: 32.4 – 285.2) and a slope parameter of 1.3 (95% CI: 0.9 – 1.8).

Figure 4a - put fold difference, in addition to the p value (e.g., as $x1.2$ ($p=xxx$)). Fig 4b - using different styles of model fit will help folks with color blindness to tell difference.

We have added the fold difference of the relative risk to the caption and the model fit to the vaccine data is now a dashed line.

Supplement, page 6 - why's is W larger than B (put specific numbers!)? Is there an intuitive explanation of this?

We have added an intuitive explanation of why W is larger than B to the Supplementary Materials and Methods.

Supp. Material: This means that the variation between bootstraps is very small compared to the uncertainty of the dose-response curve parameters for each bootstrapped data set, i.e. the uncertainty in the x-position (the dose) appears to be small compared to the error in the y-position (the efficacy estimate). Due to low numbers of events, the uncertainty in the efficacy estimate is large in some cases (see the confidence intervals for the efficacy in Figure 1 and Figure 2).

Table S5 - please list best fit parameters here. Also, make a column of delta AIC, so model comparison is made easier.

Amended.

Figure S2 and Table S5 - I found it interesting that the threshold model I suggested in my review fits the data nearly with identical quality as the main model (it is second best). I think this should be discussed some more - e.g., that you cannot tell which model is the right one. Why do you think threshold model fits the data so well? Adding AIC or delta values to each

panel in Fig S2 would be useful, perhaps along with estimated model parameters. You have space in top left corners.

We have now added Δ AIC and estimated parameter values to Table S5 and the AICs to the top left corners of the panels in Figure S2, as suggested by the reviewer.

Fig S5 - put AIC values (or better delta AIC) for each panel/model fit.

We have added AIC values to each panel in Figure S5 and Δ AIC values to Table S9.

Reviewer #2 (Remarks to the Author):

The authors have responded to all my comments or provided satisfactory rebuttals as to why this is not possible. The revised manuscript taking into account all reviewers' comments is much improved. I look forward to seeing this manuscript in print.

Reviewer #3 (Remarks to the Author):

The authors have adequately addressed my comments.

A small number of remaining notes:

1. Figure 4B appears to remain unchanged, whereas the rebuttal letter states that it has been truncated "further to the right on the x-axis".

Thank you for noting the edited version of the figure was missing from the last revision. We apologise for this oversight, we have now added the corrected figure with truncated x-axis (Figure 4B).

2. The section dealing with potential differences in antibody-mediated Fc effector functions needs some correction.

What has been modified on some of the antibodies are the Fc domains, not the Fc receptors (these are expressed on effector cells).

In addition, while adintrevimab, tixagevimab, and cilgavimab do indeed have Fc domain mutations that increase half-life through enhanced FcRn binding, it is rather unlikely that these mutations will interfere much with Fc effector functions. For adintrevimab, Fc effector functions should be mostly unaffected compared to a regular IgG antibody ("LA" mutation; see also data in the Schmidt et al. paper). For tixagevimab/cilgavimab, mutations that reduced Fc effector functions were specifically introduced ("TM" mutations) and are distinct from the mutations that enhance half-life ("YTE" mutations).

We thank the reviewer for these corrections, we have now captured these details more accurately in this discussion point:

Line 318: The difference in protection at a given neutralization titer between vaccination and monoclonal antibody therapy may be due to the additional benefit in vaccinees of a polyclonal antibody response, other non-neutralizing functions of antibodies, recall of immune memory, and/or other cellular immune responses. In particular, adintrevimab, tixagevimab and cilgavimab all have modified Fc domains to increase the antibody half-life^{13, 35}, and tixagevimab and cilgavimab have additional mutations to reduce Fc related antibody functions. These functions may contribute to the estimated trend towards higher protection for vaccines at high neutralizing titers. While our analysis has shown that neutralizing antibodies alone are sufficient to provide a high level of protection from COVID-19 at the neutralisation titres induced by vaccination, it is not possible to conclude from this analysis that neutralizing antibodies are necessary for protection. Also, since adintrevimab, casirivimab and imdevimab have seemingly intact Fc receptor functions, differences in efficacy between these products may arise from other non-neutralizing functions that we have not directly considered here. We note that evidence in animal models supports the findings that neutralizing antibodies mediate protective immunity³⁶, with some showing an additional benefit of Fc-receptor function³⁷.

3. Lines 284/285: Instead of writing “approx. 95-fold”, I would suggest to write “approx. 100-fold”.

Amended.